# Chemical Heterogeneities in the Mantle: Progress Towards a General Quantitative Description (revision II)

Massimiliano Tirone[1]

[1]No affiliation

**Correspondence:** M. Tirone (max.tirone@gmail.com)

**Abstract.** Chemical equilibration between two different assemblages (peridotite-type and gabbro/eclogite-type) has been determined using basic thermodynamic principles and certain constraints and assumptions regarding mass and reactions exchange. When the whole system (defined by the sum of the two sub-systems) is in chemical equilibrium the two assemblages will not be homogenized but they will preserve distinctive chemical and mineralogical differences. Furthermore, the mass transfer between the two sub-systems defines two petrological assemblages that separately are also in local thermodynamic equilibrium. In addition, when two assemblages previously equilibrated as a whole in a certain initial mass ratio are held together assuming a different proportion, no mass transfer occurs and the two sub-systems remain unmodified.

By modeling the chemical equilibration results of several systems of variable initial size and different initial composition it is possible to provide a quantitative framework to determine the chemical and petrological evolution of two assemblages from an initial state, in which the two are separately in chemical equilibrium, to a state of equilibration of the whole system. Assuming that the local Gibbs energy variation follows a simple transport model with an energy source at the interface, a complete petrological description of the two systems can be determined over time and space. Since there are no data to constrain the kinetic of the processes involved, the temporal and spatial scale is arbitrary. The evolution model should be considered only a semi-empirical tool that shows how the initial assemblages evolve while preserving distinct chemical and petrological features. Nevertheless despite the necessary simplification, a 1-D model illustrates how chemical equilibration is controlled by the size of the two sub-systems. By increasing the initial size of the first assemblage (peridotite-like), the compositional differences between the initial and the final equilibrated stage become smaller, while on the eclogite-type side the differences tend to be larger. A simplified 2-D dynamic model in which one of the two sub-systems is allowed to move with a prescribed velocity, shows that after an initial transient state, the moving sub-system tends to preserve its original composition defined at the influx side. The composition of the static sub-system instead progressively diverges from the composition defining the starting assemblage. The observation appears to be consistent for various initial proportions of the two assemblages, which simplify somehow the development of potential tools for predicting the chemical equilibration process from real data and geodynamic applications.

Four animation files and the data files of three 1-D and two 2-D numerical models are available following the instructions in the supplementary material.

## 1 Introduction

Our understanding of the Earth and planetary interiors is based on the underlying assumption that thermodynamic equilibrium is effectively achieved on a certain level, which means that the system under consideration is in thermal, mechanical and chem-
ical equilibrium within a certain spatial and temporal domain. Although this may appear to be just a formal definition, it affects the significance of geophysical, petrological and geochemical interpretations of the Earth's interior. While the assumption of thermodynamic equilibrium is not necessarily incorrect, the major uncertainty is the size of the domain on which the assumption is expected to be valid.

The Earth and planetary interior as a whole could be defined to be in mechanical equilibrium when the effect of the grav-
itational field is compensated, within a close limit, by a pressure gradient (for simplicity variations of viscous forces are neglected). Even when this is effectively the internal state (one example could be perhaps the interior of Mars), thermodynamic equilibrium most likely is not achieved because it requires also thermal equilibrium (i.e. uniform temperature) and chemical equilibrium (for possible definitions of chemical equilibrium see for example Prigogine and Defay, 1954; Denbigh, 1971; Smith and Missen, 1991; Kondepudi and Prigogine, 1998). On a smaller scale instead, local thermodynamic equilibrium could
be a reasonable approximation. If the system is small enough, the effect of the gravitational field is negligible and a condition close to mechanical equilibrium is achieved by the near balance between the gravitational force and pressure (locally both density and pressure are effectively uniform and viscous forces are neglected for simplicity). Clearly a perfect balance will lead to static equilibrium. On the other end dynamic equilibrium makes harder for chemical and thermal equilibrium to be maintained. In studies of planetary solid bodies it is often reasonable to assume dynamic equilibrium close to a quasi-static condition in
which the forces balance is close but not exactly zero. At a smaller scale it is then easier to consider that the temperature is also nearly uniform. The main uncertainty remains the chemical equilibrium condition. On a planetary scale, whether the size of system under investigation is defined to be on the order of hundreds of meters or few kilometers, it has little effect on the variation of the gravity force and in most cases on the temperature gradient. But for chemical exchanges, the difference could lead to a significant variation of the extent of the equilibration process. For the Earth's mantle in particular this is the
case because it is generally considered to be chemically heterogeneous. The topic has been debated for some time (Kellogg, 1992; Poirier, 2000; Schubert et al., 2001; van Keken et al., 2002; Helffrich, 2006) and large scale geodynamic models to study chemical heterogeneities in the Earth's mantle have been refined over the years (Gurnis and Davies, 1986; Ricard et al., 1993; Christensen and Hofmann, 1994; Walzer and Hendel, 1999; Tackley and Xie, 2002; Zhong, 2006; Huang and Davies, 2007; Brandenburg et al., 2008; Li and al., 2014; Ballmer et al., 2015, 2017). Geochemical (van Keken and Ballentine, 1998;
van Keken et al., 2002; Kogiso et al., 2004; Blusztajn et al., 2014; Iwamori and Nakamura, 2014; Mundl et al., 2017) and geophysical (van der Hilst et al., 1997; Trampert et al., 2004; Tommasi and Vauchez, 2015; Zhao et al., 2015; Tesoniero et al., 2016) data essentially support the idea that the mantle develops and preserves chemically heterogeneities through the Earth's history. Even though all the interpretations of the mantle structure are based on the assumption of local thermodynamic equilib-

rium, the scale of chemical equilibration has never been investigated in much detail. An early study (Hofmann and Hart, 1978) suggested that chemical equilibrium cannot be achieved over a geological time, even for relatively small systems (kilometer scale), hence it must preserve chemical heterogeneities on the same scale. The conclusion was inferred based on volume diffusion data of Sr in olivine at $1000^oC$. At that time the assessment was very reasonable, albeit the generalization was perhaps an oversimplification of a complex multiphase multicomponent problem. In any case, significant progress in the experimental methodology to acquire kinetic data and better understanding of the mechanisms involved suggest that the above conclusion should be at least reconsidered. Based on the aforementioned study, the only mechanism that was assumed to have some influence on partially homogenizing the mantle was mechanical thinning/mixing by viscous deformation (Kellogg and Turcotte, 1987). In addition very limited experimental data on specific chemical reactions relevant to mantle minerals (Rubie and Ross II, 1994; Milke et al., 2007; Ozawa et al., 2009; Gardés et al., 2011; Nishi et al., 2011; Dobson and Mariani, 2014) came short to set the groundwork for a general re-interpretation of chemical heterogeneities in the mantle.

Perhaps a common misconception is that chemical equilibrium between two lithologies implies chemical homogenization. In other words, if the mantle is heterogeneous, chemical equilibration must have not been effective. This is not necessarily true. A simple example may explain this point. If we considering for example the reaction between quarz and periclase to form variable amount of forsterite and enstatite: $MgO + nSiO_2 \Rightarrow (1-n)Mg_2SiO_4 + (2n-1)MgSiO_3$, at equilibrium, homogenization would require the formation of a bimineralic single layer made of a mixture of enstatite and forsterite crystals. However experimental studies (e.g. Gardés et al., 2011) have shown instead the formation of two separate monomineralic layers, one made of policrystalline enstatite and the other one made of forsterite.

In summary there are still unanswered questions regarding the chemical evolution of the Earth's mantle, for example, at what spatial and temporal scale we can reasonably assume that a petrological system is at least close to chemical equilibrium? and how does it evolve petrologically and mineralogically?

This study expands a previous contribution that aimed to provide an initial procedure to determine the chemical equilibration between two lithologies (Tirone et al., 2015). The problem was exemplified in a illustration (figure 1 in Tirone et al. (2015)). Because certain assumptions need to be made, the heuristic solution, further developed here, is perhaps less rigorous than other approaches based on diffusion kinetics that were applied mainly for contact metamorphism problems (Fisher, 1973; Joesten, 1977; Nishiyama, 1983; Markl et al., 1998). However the advantage is that it is relatively easy to generalize, and it leads towards a possible integration with large scale geodynamic numerical models while still allowing for a comparison with real petrological data. At the same time it should be clear that to validate this model approach and to coinstrain the extent of the chemical equilibration process, experimental data should be acquired on the petrological systems investigated here and in the previous study.

The following section (section 2) outlines the revised procedure to determine the two petrological assemblages forming together a system in chemical equilibrium. The revision involves the method used to determine the composition of the two assemblages when they are in equilibrium together, the database of the thermodynamic properties involved and the number of oxides considered in the bulk composition. In addition since the solids are non-ideal solid mixtures (in the previous study all mixtures were ideal), the chemical equilibration requires that the chemical potential of the same components in the two

assemblages must be the same (Prigogine and Defay, 1954; Denbigh, 1971). The method is still semi-general in the sense that a similar approach can be used for different initial lithologies with different compositions, however some assumptions and certain specific restrictions should be modified depending on the problem. The simplified system discussed in the following sections assumes on one side a peridotite-like assemblage, and a gabbro/eclogite on the other side. Both are considered at a

fixed pressure and temperature (40 kbar and $1200^{o}C$) and their composition is defined by nine oxides. The general idea is to conceptually describe the proxy for a generic section of the mantle and a portion of a subducting slab. A more general scheme that allows for variations of the pressure and temperature should be considered in future studies. The results of the equilibration method applied to 43 different systems are presented in section 2.1. The parameterization of the relevant information that can be used for various applications is discussed in section 2.2. Section 3 presents the first application of a 1-D numerical model

applied to pairs of assemblages in variable initial proportions to determine the evolution over time towards a state of equilibration for the whole system. The next section (4) illustrates the results of few simple 2-D dynamic models that assume chemical and mass exchange when one side moves at a prescribed velocity while the other side remains fixed in space. These simple models only serve the purpose to illustrate how distinct mineralogical and petrological features are preserved after chemical equilibration has been reached.

All the necessary thermodynamic computations are performed in this study with the program AlphaMELTS (Smith and Asimow, 2005), which is based on the thermodynamic modelization of Ghiorso and Sack (1995); Ghiorso et al. (2002) for the melt phase, the mixture properties of the solid and certain end-member solids. The thermodynamic properties of most of the end-member solid phases are derived from an earlier work (Berman, 1988). Even though melt is not present at the (P,T,X) conditions considered in this study, and other thermodynamic models are also available (Saxena, 1996; Stixrude and Lithgow-Bertelloni,

2005; Piazzoni et al., 2007; de Capitani and Petrakakis, 2010; Holland and Powell, 2011; Duesterhoeft and de Capitani, 2013), AlphaMELTS proved to be a versatile tool to illustrate the method described in this work. It also allows for a seamless transition to potential future investigations in which it would be possible to study the melt products of two equilibrated, or partially equilibrated assemblages when the P,T conditions are varied.

## 2   Modeling Chemical Equilibration Between Two Assemblages

This section describes in some details the procedure to determine the transformations of two assemblages after they are put in contact and the system as a whole reaches a condition of chemical equilibrium. The bulk composition is described by nine oxides ($SiO_2$, $TiO_2$, $Al_2O_3$, $Fe_2O_3$, $Cr_2O_3$, $FeO$, $MgO$, $CaO$, $Na_2O$). Retaining the input format of the AlphaMELTS program, the bulk composition is given in grams. Pressure and temperature are defined at the beginning of the process and they are kept constant. Water (thermodynamic phase) is not considered simply because the mobility of a fluid phase (or melt) cannot be easily quantified and incorporated in the model. Three independent equilibrium assemblages are retrieved using

AlphaMELTS. These are standard equilibrium computations which consist of solving a constrained minimization of the Gibbs free energy (van Zeggeren and Storey, 1970; Ghiorso, 1985; Smith and Missen, 1991). The first two equilibrations involve the bulk compositions of the two assemblages separately. The third one is performed assuming a weighted average of the

bulk composition of the two assemblages in a predefined proportion, for example 1:1, 5:1 or 100:1, also expressed as f:1 where f=1,5,100 (peridotite : gabbro/eclogite). This third computation applies to a whole system in which the two assemblages are now considered sub-systems. The variable proportion essentially allows to put increasingly larger portions of the sub-system mantle in contact with the sub-system gabbro/eclogite using the factor $f$ to indicate the relative "size" or mass of

material involved. By using AlphaMELTS the mineralogical abundance and composition in moles is retrieved from the file `phase_main_tbl.txt`, while the chemical potential for each mineral component in the solid mixture is retrieved from the thermodynamic output file (option 15 in the AlphaMELTS program). Knowing all the minerals components involved, an independent set of chemical reactions can be easily found (Smith and Missen, 1991). For the problem in hand, the list of minerals and abbreviations are reported in table 1, and the set of independent reactions are listed in table 2.

Given the above information, the next step is to determine the bulk composition and the mineralogical assemblages of the two sub-systems after they have been put together and equilibration of the whole system has been reached. For this problem the initial amount of moles $n$ of mineral components $i$ in the two assemblages is allowed to vary ($\Delta n_i$), provided that certain constraints are met. The set of constraints can be broadly defined in two categories. The first group consist of relations that are based on general mass, chemical or thermodynamic principles. The second set of constraints are based on certain reasonable

assumptions that should be verified by future experimental studies.

The first and most straightforward set of constraints requires that the sum of the moles in the two assemblages should be equal to the moles of the whole system:

$$\frac{f\left[n_i(A_0) + \Delta n_i(A)\right] + \left[n_i(B_0) + \Delta n_i(B)\right] - (f+1)n_i(W)}{(f+1)n_i(W)} = 0 \tag{1}$$

where $n_i(A_0)$ represents the initial number of moles of the mineral component in the first assemblage (A) in equilibrium before it is put in contact with the second assemblage (B). A similar definition applies to $n_i(B_0)$. $\Delta n_i(A)$ and $\Delta n_i(B)$ are the variations of the number of moles after the two assemblages are held together and $n_i(W)$ is the number of moles of the component in the whole assemblage $(A + B)$. The size of the whole assemblage is defined by $f + 1$ where $f$ refers to the size of the first assemblage.

Another set of constraints imposes the condition of local chemical equilibrium (Prigogine and Defay, 1954; Denbigh, 1971; Kondepudi and Prigogine, 1998) by requiring that the chemical potentials of the mineral components in the two sub-systems cannot differ from the chemical potentials found from the equilibrium computation for the whole assemblage ($W$):

$$\left|\frac{\mu_i(A) - \mu_i(W)}{\mu_i(W)}\right|^2 + \left|\frac{\mu_i(B) - \mu_i(W)}{\mu_i(W)}\right|^2 = 0 \tag{2}$$

where $\mu_i(A)$ is the chemical potential of the mineral component in the assemblage $A$ whose number of moles is $n_i(A) = n_i(A_0) + \Delta n_i(A)$, and a similar expression for the second assemblage $B$.

Another constraint is given by the sum of the Gibbs free energy of the two sub-systems that should be equal to the total Gibbs free energy of the whole system:

$$\left(\frac{fG(A) + G(B) - (f+1)G(W)}{(f+1)G(W)}\right)^2 = 0 \tag{3}$$

where $G(A) = \sum_i n_i(A)\mu_i(A)$ and similar equations for $B$ and $W$.

The list of reactions in table 2 allows to define a new set of equations which relates the extent of the reaction $\xi_r$ with the changes of the moles of the mineral components (Prigogine and Defay, 1954; Kondepudi and Prigogine, 1998). Consider for example the garnet component almandine (Alm) which appears in reaction (T-1), (T-3), (T-10), (T-12), (T-13), (T-14), (T-15) and (T-16), the following relation can be established:

$$f\Delta n_{Alm}(A) + \Delta n_{Alm}(B) \quad +1\,\xi_{(T-1)} \quad +1\,\xi_{(T-3)} + 1\,\xi_{(T-10)} + 1\,\xi_{(T-12)} + 1\,\xi_{(T-13)} \tag{4}$$
$$+1\,\xi_{(T-14)} \quad +1\,\xi_{(T-15)} - 1\,\xi_{(T-16)} = 0$$

where all the extent of the reactions are considered to be potential new variables. However not necessarily all the $\xi_r$ should be treated as unknowns. This can be explained by inspecting for example table 3, which provides the input data and the results of the equilibrium modeling of on of the study cases, in particular the one that assumes an initial proportion 1:1 (f=1). The second and third column on the upper side of the table report the input bulk composition on the two sides. The second and fifth column on the lower part of the table show the results of the thermodynamic equilibrium calculation applied separately to the two sub-systems. The last column shows the results for the whole system $W$. This last column indicates for example that orthopyroxene is not present at equilibrium in the whole assemblage. Considering the reactions in table 2 and the data in table 3, the En component in orthopyroxene appears only in reaction T-2, and since no OEn is present on the $B$ side, the mole change in $A$ can be locked ($\Delta n_{OEn}(A) = -0.0700777$). Therefore $\xi_{(T-2)}$ is fixed to -0.0700777. The same is also true for $\xi_{(T-3)}$ which is uniquely coupled to $\Delta n_{OEss}(A)$, furthermore $\xi_{(T-4)}$ coupled to $\Delta n_{OHd}(A)$, also $\xi_{(T-11)}$ coupled to $-\Delta n_{OJd}(A)$, and finally $\xi_{(T-17)}$ fixed by $\Delta n_{Coe}(B)$.

For the problem in hand the above set of relations does not allow to uniquely define the changes of the moles of the mineral components in the two sub-systems. Therefore additional relations based on some reasonable assumptions have been added to the solution method. Future experimental studies will need to verify the level of accuracy of such assumptions. Certain constraints on the mass exchange can be imposed by comparing the equilibrium mineral assemblage of the whole system ($W$) with the initial equilibrium assemblages in $A_0$ and $B_0$. For example table 3 shows that olivine is present in the whole assemblage $W$. However initially olivine is only located in sub-system $A_0$. Therefore rather than forming a complete new mineral in $B$, the assumption is that the moles of fayalite (Fa), monticellite (Mtc) and forsterite (Fo) will change only in sub-system $A$ to comply with the composition found for the whole assemblage $W$. Following this reasoning the changes in the two sub-systems could be set as: $\Delta n_{Fa}(A) = 0.0008090$, $\Delta n_{Mtc}(A) = -0.0000555$ and $\Delta n_{Fo}(A) = -0.0726300$ and $\Delta n_{Fa}(B) = \Delta n_{Mtc}(B) = \Delta n_{Fo}(B) = 0$. In this particular case the same assumption is also applicable to the orthopyroxene components. It is clear that starting with different bulk compositions or proportions or (T,P) conditions, alternative assemblages may be formed, therefore different conditions may apply, but the argument on which the assumption is based should be similar. Additional constraints based on further assumptions can be considered. For example, garnet appears on both sides $A_0$ and $B_0$. The components pyrope (Prp) and grossular (Grs) contribute only to two reactions, (T-1) and (T-12), and in both cases the reactions involve only olivine components which have been fixed in sub-system $A$, as previously discussed. The assumption that is made here is that the change of the moles of the garnet components in sub-system $B$ will be minimal because no olivine

is available in this sub-system. Therefore the following relation is applied:

$$min \left( \frac{\Delta n_{Prp}(B)}{n_{Prp}(B_0)} \right)^2 \tag{5}$$

and similar relations can be also imposed to the other garnet components, Alm and Grs. The same argument can be applied to the clinopyroxene and spinel components. For example the spinel component hercynite (Hc) appears only in reaction (T-13), which involves olivine and orthopyroxene components (Fa, ODi) located in sub-system $A$, and the garnet component Alm which has been already defined by the previous assumption.

The overall procedure is implemented with the use of Minuit (James, 1994), a program that is capable of performing a minimization of multi-parameter functions. Convergence is obtained making several calls of the Simplex and Migrad minimizers (James, 1994). The procedure is repeated with different initial values for the parameters $\Delta n_i(A)$, $\Delta n_i(B)$ and $\xi_r$ to confirm that a unique global minimum has been found.

## 2.1 Results of the Chemical Equilibrium Model Between Two Assemblages

This procedure described in the previous section has been applied to 43 different cases, varying the proportion of the two sub-systems from 1:1 to 1000:1 and considering different, but related, initial compositions. The initial bulk composition and the proportion factor $f$ of the two sub-systems for all the 43 cases are included in a table available in the supplementary material. For example the initial compositions for $A_0$ and $B_0$ applied to case #11 are taken from table 4 (column $A*$) and from table 3 (column $B_0$), both tables discussed in this section. Tables 3-7 report the results of the procedure discussed in the previous section for few cases. Table 3 was briefly introduced earlier to show the initial bulk composition of the two sub-systems (upper portion of the table), the initial equilibrium assemblages and the mole changes after the chemical equilibration (lower part of the table). The table also includes the bulk composition in the two sub-systems after the chemical equilibration procedure is completed (upper part, column 5 and 6). These bulk compositions are calculated from the mole abundance of the mineral components shown in the lower part (columns 4 and 7). The total mass of the sub-systems is reported as well. Note that negative abundance of certain mineral components is permissible according to the thermodynamic model developed by Ghiorso (Ghiorso and Carmichael, 1980; Ghiorso, 2013) as long as the related oxides bulk abundance is greater than zero.

In the example shown in table 3 there is a significant mass transfer from $B$ to $A$: mass($A_0$)=100, mass($A$)=146.36 and mass($B_0$)=100, mass($B$)=53.64 (grams). The table also includes the total Gibbs energy for the sub-systems, before and after the equilibration of the whole system which are computed from the output of the program AlphaMELTS after combining the moles of the components and the relative chemical potentials. The total Gibbs free energy is relevant for the parameterization discussed in the next section. Table 4 is a summary of a further analysis aiming to investigate whether there is any pattern in the compositions of the two sub-systems. The bulk compositions in the upper portion of the table ($A*$, $B*$) are obtained by normalizing the oxides in $A$ and $B$ (upper part, column 5 and 6 of table 3) to a total mass of 100 grams. For example $SiO_2$ in $A*$ from table 4 (47.434) is $100\times(SiO_2$ in $A$)/(sum of oxides in $A$) from table 3, which is equal to $100\times69.428/146.367$. The normalized oxides ($A*$, $B*$) represent the mass of the components in grams when the total mass is 100 grams, which is obviously also equivalent to the weight % of the components. These bulk compositions can

be used for two new Gibbs free energy minimizations, one for each of the two sub-systems, to retrieve the correspondent equilibrium assemblages separately. The interesting observation that can be made following the summary in the lower part of table 4, is that the abundance of the mineral components remains unmodified after scaling the results for the total mass of the system. For example using the data from table 3, the proportion relation: $n_{alm}(A) : 146.347 = n_{alm}(A*) : 100$ gives

$n_{alm}(A*) = n_{alm}(A) \times 100/146.347 = 0.01453 \times 0.6833 = 0.009928$ which is remarkably close to the moles of almandine found from the separate equilibration calculation reported in table 4, $n_{alm}(A*) = 0.0099353$. In other words the scaling factor used to define the input oxide bulk composition can be also applied to the equilibrium mineral assemblage.

Based on this observation, some equilibration models have been carried out considering at least one of the initial composition from a previous model (e.g. $A*$ from a previous equilibration model $\Rightarrow$ input for a new model $A_0$ or alternatively $B* \Rightarrow B_0$),

while for the other sub-system the initial bulk composition from table 3 is used again. A special case is the one shown in table 5 in which both $A_0$ and $B_0$ are taken from the equilibrated and normalized data of the previous model, $A*$ and $B*$, reported in table 4. If the proportion in the new model remains the same, 1:1, then clearly no compositional changes are expected since the whole system is already in equilibrium. If the proportion is changed, for example to 5:1 ($f = 5$), the bulk composition of the whole system is different from the bulk composition of the whole system with 1:1 proportion and the assemblages in the two

sub-systems may not remain unmodified after equilibration. However this does not appear to be the case, as shown in table 5, where $\Delta n_i(A)$ and $\Delta n_i(B)$ are very small. The results suggest that the moles of the mineral components remain unchanged. A more general case with $f = 5$ is presented in table 6. The model is essentially the same shown in table 3, but with proportion of the two initial sub-systems set to 5:1. As expected the results of the equilibration process are different from the results starting with an initial proportion 1:1 (table 3). For example with 1:1, $n_{alm}(A) = 0.01453$, while with 5:1, $n_{alm}(A)/5 = 0.00737$.

The question is whether the observation made for the first studied case with proportion 1:1 can be generalized. In particular the observation that the minerals abundance in the two sub-systems from the equilibration procedure of the whole system is equivalent to the one that is obtained from two separate equilibration computations using the normalized bulk compositions $A*$ and $B*$. Indeed it appears that the same conclusion can be made for the model with 5:1 initial proportion (table 7). The number of moles of the almandine component is $(n_{alm}(A)/5) \times 100/110.064 = 0.006698$ (table 6) which can be compared

with $n_{alm}(A*) = 0.006695$ from table 7. The similarity has been also observed for all the other models with $f$ ranging from 1 to 1000.

## 2.2   Parameterization of the Equilibrium Model Results for Applications

While interesting observations have been made about the mineralogical assemblages in the two sub-systems after chemical equilibration, it is still unclear how this type of model can be applied for studies on the chemical evolution of the mantle.

Figure 1 summarizes the relevant data that allows to determine the bulk composition and the mineralogical assemblage in the two sub-systems after the chemical equilibration process is completed.

The key quantity is the normalized Gibbs energy of the two sub-systems after they have been equilibrated, $G(A*)$ and $G(B*)$. The normalized Gibbs energy for an unspecified sub-system (either $A*$ or $B*$) is defined by the symbol $G(*)$. The quantity can be computed from the AlphaMELTS output after the Gibbs free energy minimization is applied to $A*$ or $B*$, or it can be

simply obtained by scaling G(A) or G(B). Panel 1-A) shows the relation between the ratio $G(A*)/G(B*)$ and $G(B*)$ which will be used later to define $G(*)$ at the interface between the two assemblages. The data in the figure for the 43 models have been fitted using a Chebyshev polynomials (Press et al., 1997). By knowing $G(*)$, it is possible to retrieve the abundance of all the oxides defining the bulk composition normalized to 100 grams. An example is shown in panels 1-B) and 1-C) which illustrate the data points for $MgO$ in $(A*)$ and $(B*)$ in the 43 study models and the fitting of the points using Chebyshev polynomials.

The mass transfer between the two sub-systems can be related to the total Gibbs free energy variation in each of the two sub-systems $G(A)$ and $G(B)$. The two relations are almost linear, as shown in panel 1-D). For practical applications, once a relation is found between $G$ and the normalized $G(*)$, then the mass transfer can be quantified. Panel 1-E) of figure 1 shows the data points and the data fitting with the Chebyshev polynomial of the function $G(B)[G(B*) - G(B_0)]$ versus $[G(B*) - G(B_0)]$. More details on the use of the fitting polynomial functions are provided in the next section.

## 3   Application to the evolution of a 1-D Static Model with Variable Extension

The chemical and petrological evolution of two assemblages can be investigated with a 1-D numerical model, assuming that the two sub-systems remain always in contact and they are not mobile. The problem is assumed to follow a simple conduction/diffusion couple-type model with variable size for the local variation of $G(*)$ which can be expressed by the following equation for each sub-system:

$$\frac{\partial G(*)}{\partial t} = S(*)\frac{\partial^2 G(*)}{\partial d_x(*)^2} \tag{6}$$

where $S(*)$ is a scaling factor and $G(*)$ and $S(*)$ refers to either $A*$ or $B*$. Time $t$, distance $d_x(*)$ and the scaling factor $S(*)$ have no specific units since we have no knowledge of the kinetic of the processes involved. At the moment these quantities are set according to arbitrary units, S(A*) and S(B*) are set to 1, while $t$, $d_x(A*)$ and $d_x(B*)$ have different values depending on the numerical simulation. It should be clear that the dynamic model provides only a semi-empirical quantitative description of a complex process. The main purpose is to illustrate the general concept and to show that the two assemblages could develop distinct regions evolving towards the condition of chemical equilibrium, while far from the interface area the initial compositions can be preserved for a certain amount of time. The detailed description on how the two sub-systems will eventually reach chemical equilibration is beyond the scope of this study.

The numerical solution with grid spacing $\Delta d_x(*)$, uniform on both sides, is obtained using the well-known Crank-Nichols method (Tannehill et al., 1997). At the interface (defined by the symbol $if$) the polynomial of the function shown in panel 1-A) of figure 1 is used together with the flux conservation equation:

$$\left.\frac{\partial G(A*)}{\partial d_x(A*)}\right|_{if} = -\left.\frac{\partial G(B*)}{\partial d_x(B*)}\right|_{if} \tag{7}$$

to retrieve $G(A*)_{if}$ and $G(B*)_{if}$ assuming that $S(A*) = S(B*)$. The external boundaries defining the limits of the whole system (symbol $l$) are assumed to be of closed-type or symmetric-type. Both are obtained by the condition $G(A*)_l = G(A*)_{n_A-1}$

and $G(B*)_l = G(B*)_{n_B-1}$, where $n_A$ and $n_B$ are the total number of grid points on each side (excluding the boundary points). $G(A*)_l$ and $G(B*)_l$ define the outside boundary limits of the whole system which represent either the closed-end of the system or the middle point of two mirrored images.

To determine the mass transfer and how it affects the length of the two sub-systems, the following steps are applied. The polynomial of the relation shown in panel 1-E) of figure 1 is used at the interface point to find $G(B)_{if}$ (from the relation with $G(B*)_{if} - G(B_0)$). Defining $\Delta G = [G(B_0) - G(B)_{if}]/G(B_0)$, the length of sub-system $B$ at complete equilibrium would be $D_{x,eq}(B*) = D_x(B_0) + D_x(B_0)\Delta G$, where $D_x(B_0)$ is the total length of the sub-system at the initial time. The spatial average of $G(B*)$, defined as $G(B*)_{av}$ can be easily computed. The quantity $G(B*)_{av}$ is needed in the following relation to find the current total length of the sub-system at a particular time:

$$D_{x,t}(B*) = D_{x,eq}(B*) - [D_{x,eq}(B*) - D_x(B_0)]\frac{G(B*)_{if} - G(B*)_{av}}{G(B*)_{if} - G(B_0)} \tag{8}$$

The same change of length is applied with opposite sign on the other sub-system. The new dimensions $D_{x,t}(A*)$ and $D_{x,t}(B*)$ define also new constant grid step sizes, $\Delta_x(A*)$ and $\Delta_x(B*)$. The final operation is to re-mesh the values of $G(*)$ at the previous time step onto the new uniform spatial grid.

It is worth to mention that in the procedure outlined above here, converting the change of $G$ to the change of the total length of the sub-system is a two steps process. The first step makes use of the relation between the change of $G$ and the change of the total mass, which was illustrated in panel 1-D) of figure 1. In the next step the assumption is that the change of mass (and $G$) is proportional to the change of the total length of the sub-system.

To summarize the numerical procedure, at every time step the complete solution on both sides is obtained by solving equation 6 for $G(A*)$ and $G(B*)$ with the boundary conditions imposed for the limits of the whole system and preliminary values for the interface points. Then the interface points are updated using the polynomial function and equation 7. The total length is then rescaled to account for the mass transfer and the numerical grid size is updated. This procedure is iterated until the variation between two iterations becomes negligible (typically convergence is set by: $|G(A*)_{if}^{\#1} - G(A*)_{if}^{\#2}| + |G(B*)_{if}^{\#1} - G(B*)_{if}^{\#2}| < 1e-4$, where the labels # 1 and # 2 refer to two iterative steps).

Once convergence has been reached, the oxide abundance can be found easily using the Chebyshev polynomial parameterization in which each oxide is related to a function of $G(A*)$ or $G(B*)$ (e.g. for $MgO$ see panel 1-B) and 1-C) of figure 1). For convenience the composition is identified in wt% since the normalized oxides (*) represent the grams of the components with respect to a total mass of 100 grams. Finally, knowing temperature, pressure and the variation of the bulk oxides composition in space and time, a thermodynamic equilibrium calculation can be performed at every grid point using the program AlphaMELTS to determine the local mineralogical assemblage.

Several 1-D numerical simulations have been carried out with initial proportion ranging from 1:1 to 100:1. Some results from a test case with proportion 1:1 are shown in figure 2. Initial total length on both side is set to $D_x(A_0) = D_x(B_0) = 100$ (arbitrary units), the initial spatial grid step is $\Delta d_x(A_0) = \Delta d_x(B_0) = 1$. Time step is set to 4 (arbitrary units) and S(A*)=S(B*)=1. The initial bulk composition of the two assemblages, that separately are in complete thermodynamic equilibrium, is the same reported in table 1: $SiO_2 = 45.2$, $TiO_2 = 0.20$, $Al_2O_3 = 3.94$, $Fe_2O_3 = 0.20$, $Cr_2O_3 = 0.40$, $FeO = 8.10$, $MgO = 38.40$,

$CaO = 3.15$, $Na_2O = 0.41$ wt% (peridotite side) $SiO_2 = 48.86$, $TiO_2 = 0.37$, $Al_2O_3 = 17.72$, $Fe_2O_3 = 0.84$, $Cr_2O_3 = 0.03$, $FeO = 7.61$, $MgO = 9.10$, $CaO = 12.50$, $Na_2O = 2.97$ wt% (gabbro/eclogite side). Panel 2-A) illustrates the variation of G(*) on both sides, at the initial time (black line) and at three different times, 80, 4000 and 20000 (arbitrary units). Note the increase of the length on the $A$ side and decrease on the $B$ side. Bulk oxides abundance is also computed at every grid point. The bulk $MgO$ (wt%) is reported on panel 2-B), which shows the progressive decrease on the $A$ side while MgO increases on the $B$ side. The bulk composition can be used with the program AlphaMELTS to determine the local equilibrium assemblage. Panels 2-C) - 2-H) show the amount of the various minerals in wt% (solid lines) and the $MgO$ content in each mineral in wt% (dotted lines), with the exception of coesite in panel 2-H) ($SiO_2$). The complex mineralogical evolution during the chemical equilibration process can be studied in some detail. For example one can observe the progressive disappearance of orthopyroxene on the peridotite side and the exhaustion of coesite on the gabbro/eclogite side.

Similar results are shown in figure 3 and 4 for models with initial proportion set to 5:1 and 50:1, respectively. Differences in the numerical setup of the new test cases can be summarized as follow. For the 5:1 case: $D_x(A_0) = 500$, $D_x(B_0) = 100$, $\Delta d_x(A_0) = \Delta d_x(B_0) = 1$, time step is set to 40, for the 50:1 case: $D_x(A_0) = 5000$, $D_x(B_0) = 100$, $\Delta d_x(A_0) = 5$, $\Delta d_x(B_0) = 1$, time step is set to 800.

Few observations can be made by comparing the three simulations. For example, orthopyroxene on the peridotite side becomes more resilient and the total amount of Opx increases with the size of the initial sub-system. On the other side it appears that the $MgO$ content in garnet (pyrope component) is greater for the model with starting proportion 5:1, compared to the 1:1 case. However with initial proportion 50:1, the $MgO$ content does not seem to change any further.

The supplementary material provides a link to access the raw data (all nine oxides) for the three test cases with initial proportion 1:1, 5:1 and 50:1. In addition two animations (1:1 and 5:1 cases) should help to visualize the evolution of the numerical models over time.

## 4 Application to the Evolution of a 2-D Model with One Dynamic Assemblage and Variable Extension

A 2-D numerical model makes possible to study cases in which at least one of the two assemblages becomes mobile. The simplest design explored in this section, considers a rectangular box with a vertical interface dividing the two sub-systems. The dynamic condition is simply enforced in the model by assuming that one of the two assemblages moves downwards with a certain velocity, replaced by new material entering from the top side, while the other assemblage remains fixed in the initial spatial frame. The whole system evolves over time following the same principles introduced in the previous section. The numerical solution of the 2-D model is approached at every time step in two stages. In the first stage the following equation is applied to both sub-systems:

$$\frac{\partial G(*)}{\partial t} = S_x(*)\frac{\partial^2 G(*)}{\partial d_x(*)^2} + S_y(*)\frac{\partial^2 G(*)}{\partial d_y^2} \tag{9}$$

where $d_x(*)$ is the general spacing in the x-direction representing either $d_x(A*)$ or $d_x(B*)$ and the vertical spacing $d_y$ is assumed to be the same on both sides. This equation is solved numerically using the alternating-direction implicit method (ADI)

(Peaceman and Rachford, 1955; Douglas, Jr., 1955) which is unconditionally stable with a truncation error $O(\Delta t^2, \Delta d_x^2, \Delta d_y^2)$ (Tannehill et al., 1997). Similar to implicit methods applied for 1-D problems, the ADI method requires only the solution of a tridiagonal matrix.

The numerical procedure described in section 3 to determine $G(*)$ at the interface is also applied here to the 2-D model. The limits of the whole system opposite to the interface (left/right) are also treated similarly, assuming either a closed-type or symmetric-type boundary. For the other two boundaries (top,bottom) the zero flux condition is imposed, $G(A*)_l^{t,b} = G(A*)_{y=1,n_y}$ and $G(B*)_l^{t,b} = G(B*)_{y=1,n_y}$ where $n_y$ is the total number of grid points in the y direction (excluding the boundaries).

In the previous section a procedure was developed to account for the mass transfer between the two sub-systems. The same method is applied for the 2-D problem. The conceptual difference is that in a 2-D problem the mass change in principle should affect the area defined around a grid point. For practical purposes however in this study it only affects the length in the horizontal x-direction, hence re-meshing due to the change of mass is applied only to determine $D_{x,t}(A*)$ and $D_{x,t}(B*)$ and the two uniform grid step sizes in the x-direction, $\Delta d_x(A*)$ and $\Delta d_x(B*)$.

Up to this point the evolution of the system is not different than what was described for the 1-D case. The dynamic component is included at every time step in the second stage of the procedure. It is activated at a certain time assuming that the chosen sub-system moves downwards with a fixed pre-defined vertical velocity (y-component). The material introduced from the top side is assumed to have the same composition of the initial assemblage as defined for the 1-D models, table 1 (and the same $G(A_0)$ and $G(B_0)$ values). This is accomplished by assigning $G(A_0)$ or $G(B_0)$ at a location near the interface which is defined by the imposed velocity. Then the $G(*)$ points are also shifted according to the prescribed velocity. Values of $G(*)_y$ on the original orthogonal grid are obtained by linear interpolation of the shifted $G(*)$ points.

Oxides bulk composition is then retrieved at each grid point over time using the same polynomial functions applied for the 1-D problem. The complete mineralogical assemblage can be also computed using AlphaMELTS as part of a post-process step after the numerical simulation is completed.

Only few 2-D simulations have been performed, specifically considering the initial proportion 1:1, 5:1 and 50:1, assuming either one of the two assemblages moving downward. Figure 5 summarizes some of the results for the case 5:1(A), i.e. with moving sub-system $A$. Initial grid specifications are: $D_x(A_0) = 500$, $D_x(B_0) = 100$, $\Delta d_x(A_0) = \Delta d_x(B_0) = 2$, $D_y(A_0) = D_y(B_0) = 50$, $\Delta d_y(A_0) = \Delta d_y(B_0) = 1$ (arbitrary units). Time step is set to 16 (arbitrary units). The scaling coefficients $S_x(*)$ and $S_y(*)$ are set to 0.01 (arbitrary units). The dynamic component is activated at time=100000 with vertical velocity set to 0.00625 (arbitrary units). The figure is a snapshot of the whole system soon after sub-system $A$ has been activated downwards (time=102400). Panel 5-A) shows the variation of $G(*)$, while panel 5-B) illustrates the bulk $MgO$ distribution (wt%). The other panels, 5-C) - 5-H), present an overview of the mineralogical distribution (flood contour-type) and the $MgO$ content in each mineral phase (line contour-type), with the exception of panel 5-H) for coesite ($SiO_2$). The panels clearly illustrate the variations introduced by the mobile sub-system $A$. On the other side there is apparently no immediate effect on the assemblage $B$, however the long term effect is significant and becomes visible in a later figure (figure 7).

Figure 6 provides a similar overview for the case assuming 5:1(B) with sub-system $B$ moving downward. The same numerical

conditions described for the previous case apply for this case as well. This figure, which shows only one time-frame soon after the sub-system is mobilized, does not appear to reveal new remarkable features. However advancing the simulation, a clear effect becomes more evident near the interface. In particular changes of the chemical and mineralogical properties moving away from the top entry side are quite significant. An animation related to figure 6 is best suited to illustrate this point. This movie file and another file for the animation related to figure 5 can be downloaded following the link provided in the supplementary material. The raw data files which include all nine oxides for both simulations are also available online.

## 5   Summary of the 1-D and 2-D Models Approaching Chemical Equilibration

Figure 7 summarizes the results of all the 1-D and 2-D numerical test models when the whole system approaches or is close to chemical equilibration. In the static scenario, exemplified by the 1-D models (solid lines), by increasing the initial size of sub-system $A$, the mineralogical and compositional variations tend to be smaller (see panels 7-C) - 7-H) and enlarged view around the interface, panels 7-C2) - 7-H2)). It is the expected behavior since any change is distributed over a larger space of the sub-system. The variations of the minerals abundance in assemblage $B$ (gabbro/eclogite-type) instead remain quite independent of the initial size of sub-system $A$. However the abundance of the minerals not necessarily is the same found in the initial assemblage. In particular the amount of garnet, clinopyroxene and coesite is quite different from the amount of these minerals in the initial assemblage. This difference is rather unaffected by the initial proportion of the two assemblages, which has been varied from 1:1 (f=1) to 100:1 (f=100).

The composition of the minerals in assemblage $A$ (e.g. $MgO$ illustrated in panels 7-CC) - 7-HH)) follows a pattern similar to the minerals abundance. As the size of the initial sub-system increases, $MgO$ tends to approach the oxide amount in the initial composition. A different result is observed for the composition of the minerals in assemblage $B$. Regardless whether the mineral abundance changes or remains close to the initial amount, the oxide composition varies quite significantly and in most minerals the difference is larger when $f$ is set to higher values.

When one of the sub-systems is allowed to move (2-D models), the general observation on the long run is that the dynamic sub-system tends to preserve the assemblage that enters in the model. In this study this assemblage is set to be equal to the initial assemblage. Note that the 2-D data plotted in figure 7 refer to an horizontal section of extracted points at the middle vertical distance $D_y/2$. When sub-system $A$ is mobile (dotted lines), the behavior of assemblage $B$ is similar to the static case, with some minerals changing their initial abundance, garnet, clinopyroxene, coesite and in part spinel. In the reverse case, with $B$ set as the dynamic sub-system, the mineralogical abundance of $A$ differs from the initial assemblage (dashed lines). But unlike the static cases, no significant variations can be noted with the increase of the initial proportion.

In terms of minerals composition (e.g. $MgO$, panels 7-CC) - 7-HH) in figure 7), the dynamic sub-system preserves the composition of the entering assemblage. The immobile assemblage instead, shows a compositional variation that is larger than any change observed for the static cases. This variation remains somehow still independent of the initial proportion of the two assemblages, at least with $f = 1, 5, 50$.

Complete data for the bulk composition, which includes all nine oxides, is available for three 1-D models and two 2-D simulations following the instructions in the supplementary material.

## 6    Conclusions

The main objective of this work was to show that a chemical heterogeneous mantle does not necessarily mean that different lithologies are in chemical disequilibrium (at least not entirely).

Often geochemical and petrological interpretations of the Earth interior rely on the achievement of thermodynamic equilibrium on a certain scale. The use of phase equilibrium data and partition coefficients, for example, does imply that chemical equilibrium has been achieved and it is maintained. Curiously, while this assumption is tacitly imposed on the most convenient dimension to interpret observed data, chemical equilibration is ignored when it comes to discuss the presence or the extent of chemical heterogeneities (i.e. chemical equilibration, in this regard, is considered ineffective) (e.g. Morgan, 2001; Ito and Mahoney, 2005a, b; Strake and Bourdon, 2009; Brown and Lesher, 2014).

Geophyisical interpretations usually require to specify certain properties, such as the density for the Earth materials under consideration. For example when the density is considered representative of real rock assemblages, the system has to be sufficiently small that the gravitational force is almost completely balanced by the pressure effect (viscous forces are ignored for simplicity), effectively establishing a quasi-static or static condition. Under this condition then, thermodynamic equilibrium can be achieved when the system is also equilibrated chemically, so that petrological constraints can be applied to determine the density of the assemblage. When different lithologies are considered in geophysical applications, it is assumed that chemical equilibrium is never achieved among them, regardless of the size of the system or the temporal scale. For studies whose conclusions are based on geological processes lasting for hundreds or billion of years, such assumption should be carefully verified considering that chemical and mass exchange are always effective to a certain extent.

The results from 43 study models (section 2.1) suggest that the imposed condition of thermodynamic equilibrium for the whole system (sum of two sub-systems) defines two new assemblages that are not homogenized compositionally or mineralogically, and their equilibrated compositions are different from those in the two initial assemblages. The two new assemblages not only define a condition of chemical equilibrium for the whole system but they also represent the equilibration within each separate sub-system. In addition, mass exchange between these equilibrated assemblages does not progress any further when the initial mass proportion of the two is varied and a new equilibration model is imposed to the newly defined whole system.

The results of the study models have been condensed in a series of parameterized functions that can be used for various applications (section 2.2).

A semi-empirical quantitative forward model was also developed to describe the evolution of the chemical equilibration process in the mantle. The model has been restricted to one set of values for the pressure and temperature and one pair of bulk compositions indicative of a peridotite-type and a gabbro/eclogite-type. The gabbro/eclogite-type can be interpreted as a portion of a subduction slab. Ignoring a thin sedimentary layer, that possibly could peel off during subduction, a large portion of the slab consists also of a depleted peridotite. Three lithologies (mantle peridotite, gabbro, depleted slab peridotite) probably

can be also approached with a chemical equilibration model similar to the one presented here. However it remains to be seen whether the difference in composition with respect to the generic peridotite assumed in this study would lead to significant new results that would justify the additional modeling effort.

A priority was given here to understand the influence on the final assemblages of various initial proportions of the two sub-systems and, to a limited extent, the effect of the initial compositions. The spatial and temporal evolution necessarily assumes arbitrary units. The reason behind it is that a comprehensive approach to study chemical heterogeneities that would include time-dependent experiments and suitable models for the interpretation of the experimental results has not been developed yet. Experimental data are also necessary to validate certain assumptions that were made to model the composition of the two equilibrated assemblages (section 2).

The choice made to describe the variation of $G(*)$ using the transport model presented in section 3 and 4 may seem rather arbitrary. While details of the transition towards chemical equilibration should be investigated by experimental studies, the main point of the models in section 3 and 4 (and of this study) is to show that different lithologies can evolve while preserving distinct chemical and mineralogical features. The idea of using the concept of local Gibbs free energy variations over time and space (Kondepudi and Prigogine, 1998) to describe the chemical changes is a practical mean to simplify a problem that otherwise becomes intractable for complex systems. The choice is not a complete abstraction, it is approximately based on the consideration that the mass exchange is not governed by the compositional gradient but by the differences in the chemical potential of the various components in the various phases (e.g. Denbigh, 1971). Ultimately only extensive experimental studies could determine whether the simple evolution model for G(*) applied in this work to an heterogeneous system can be considered a reasonable approximation for describing the chemical evolution in practical geodynamic mantle models.

Two aspects of the numerical applications presented in the previous sections deserve perhaps a further consideration. The assumption made for the composition of the entering assemblage in the 2-D models perhaps should be reconsidered in future studies. The other consideration concerns the boundary condition imposed on the opposite side of the interface between the two assemblages. The assumption is that the whole system is either close to mass exchange or mirror images exist outside the boundary limits. From a geological perspective the first scenario is probably the more difficult to realize. On the other hand the possibility that periodic repetitions of the same model structure are replicated over a large portion of the mantle, if not the entire mantle, seems more reasonable. Assuming that the time scale is somehow constrained, an investigation of the temporal evolution would still require some kind of assessment of the periodic distribution of the thermodynamic system as a whole.

The 2-D simulations in which one of the assemblages is allowed to move, have shown that on the long run the mineralogical abundance and compositional variations are approximately independent of the size of the two sub-systems. This observations suggests the possibility of implementing large geodynamic models with evolving petrological systems, once the temporal and spatial scale of the chemical changes have been constrained.

At the moment the spatial and temporal variations are arbitrarily defined, but this study shows that the petrological and mineralogical changes may still be approximately quantified, at least at the (P,T) conditions that have been considered. It would be useful for example to select few bulk compositions for the two sub-systems and apply them to the dynamic equilibrium melting (DEM) and dynamic fractional melting (DFM) models that have been developed combining 1-D multiphase flow

with AlphaMELTS (Tirone and Sessing, 2017; Tirone, 2018). Perhaps even a simplified model for non-equilibrium fractional crystallization could be applied to try to reproduce observed 3-D chemical zoning in minerals and multicomponent chemical zoning in melts (Tirone et al., 2016). More in general the results should be compared with existing data on melt products and residual solids observed in various geological settings to investigate indirectly, but from a quantitative perspective, the presence

of chemical heterogeneities in the mantle. It becomes also possible to determine the variation of physical properties, such as bulk density, and relate them to certain observables, such as seismic velocities. At least on a relative scale, the effect of the compositional variations could be associated to seismic velocity variations, providing in this way another indirect evidence of heterogeneities in the mantle based on a quantitative forward description.

*Data availability.*   Supplementary material included

*Competing interests.*   No competing interests are present

*Acknowledgements.*   Thanks to Paula Antoshechkina for taking the time to answer many questions regarding the use of AlphaMELTS (version 1.7), and for fixing on-the-fly minor issues of the program. The work was carried out while visiting the Department of Mathematics and Geosciences at the University of Trieste, Italy. This study was part of a larger comprehensive project aiming to investigate chemical heterogeneities in the mantle by providing a first set of experimental data to determine the kinetics of the equilibration process, establishing a

modeling procedure and developing geodynamic numerical applications. (research proposal ID# 856505, "GEO-DIVE: Experimenting and Modeling Chemical GEO-DIVErsities in the Solid Earth", ERC-2019-SyG, ERC Synergy Program). Funding for this project was declined.

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

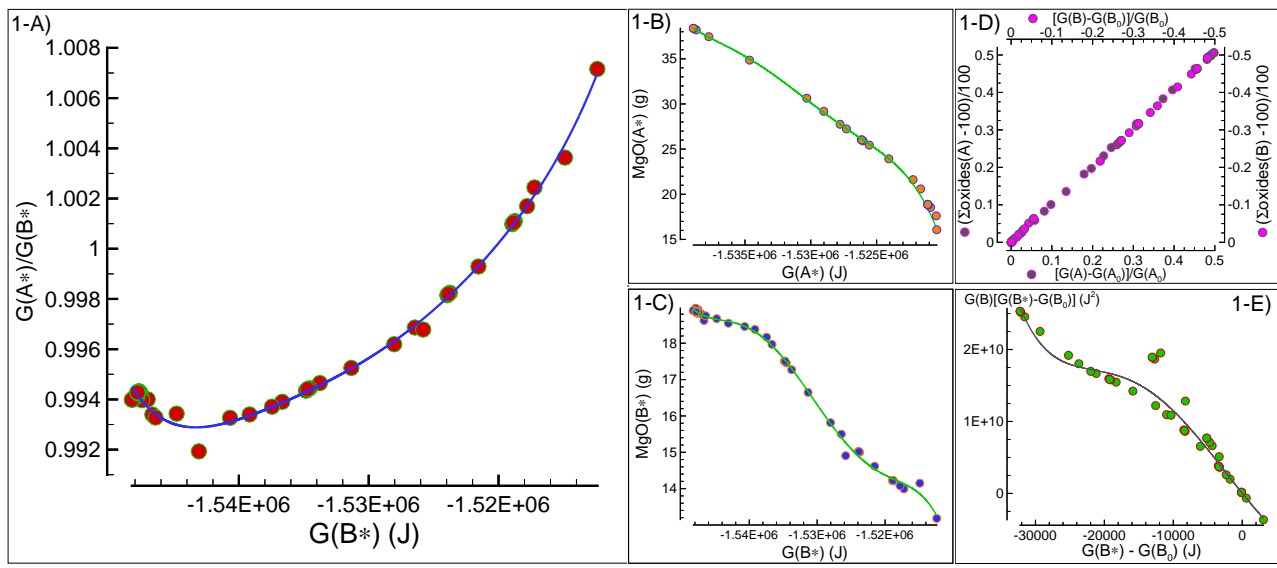

**Figure 1.** Data and relative fitting of 43 study cases that are used to develop the chemical equilibration model. Panel 1-A) relation between the ratio $G(A*)/G(B*)$ and $G(B*)$ which is applied to constrain $G(A*)$ and $G(B*)$ at the interface. Panel 1-B) and 1-C) illustrate the relation between $G(A*)$ and $G(B*)$ with $MgO$ bulk abundance. Similar relations are applied for all nine oxides defining the bulk composition. The normalized bulk abundance is intended as grams with respect to a total mass of 100 grams which is equivalent to wt%. Knowing $G(B)$, the total size of the assemblage at equilibrium can be found assuming that a) a relation between the mass change and the change of $G(B)$ is established (Panel 1-D), b) the extension of the assemblage is proportional to the mass change and it takes place along a direction perpendicular to the interface. The total length at equilibrium is then adjusted in accordance with the difference between the spatial average $G(B*)$ of the assemblage and $G(B*)$ at the interface (see the main text for a detailed explanation). The change of size of the second assemblage is also applied on the first assemblage but with opposite sign. Panel 1-E) allows to determine $G(B)$ from the relation with $G(B*)$ at the interface.

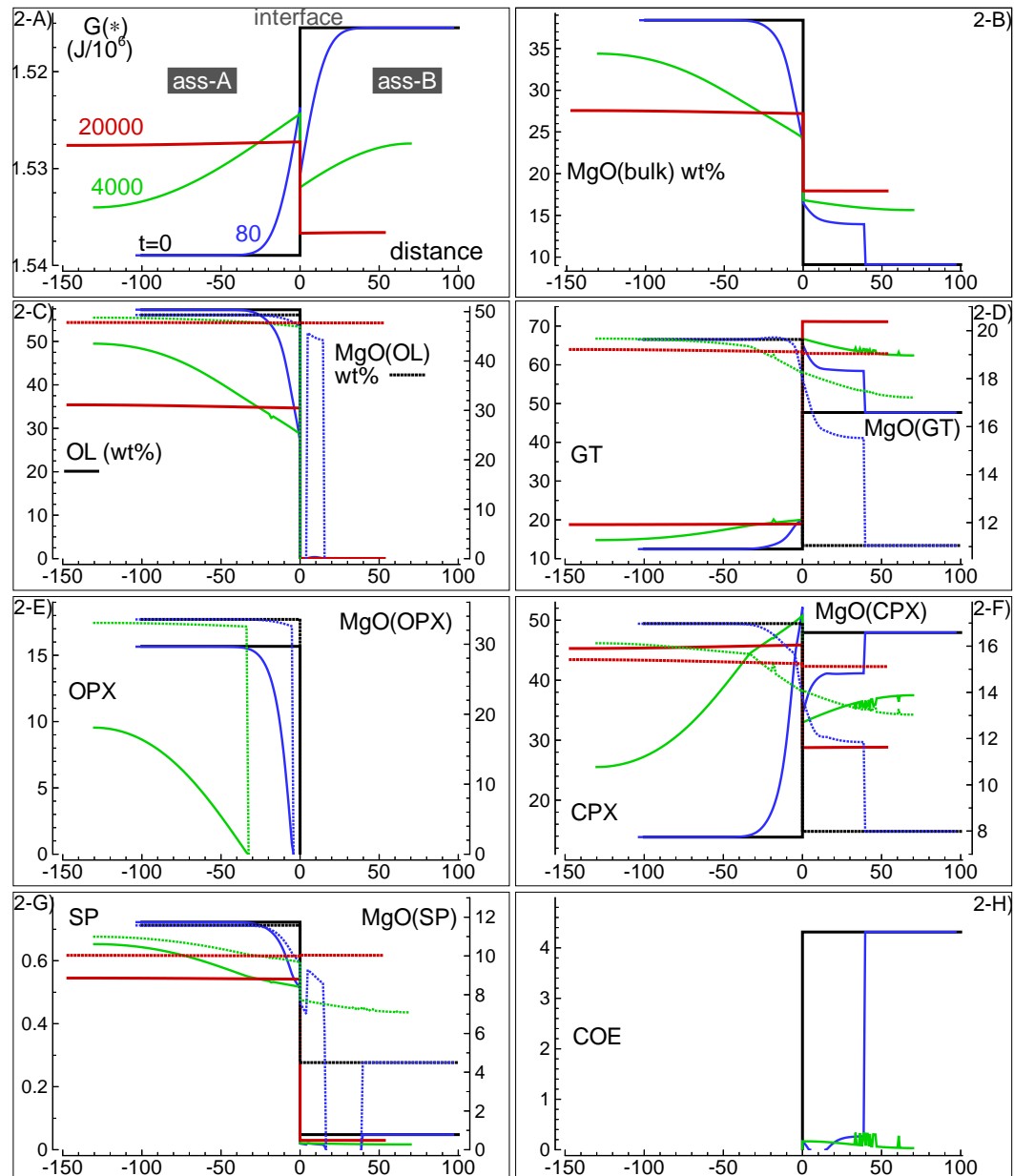

**Figure 2.** Solution of a 1-D model simulation. The initial proportion of the two assemblages is 1:1. Panel 2-A) $G(A*)$ and $G(B*)$ at three different times and at time zero when the two assemblages separately are considered in chemical equilibrium. Panel 2-B) Local bulk $MgO$ (wt%) retrieved from the relation with $G(*)$. All the other oxides are retrieved with similar relations. The units of the oxides is wt% which is equivalent to the mass of the components in grams with respect to a total mass of 100 grams. Panels 2-C) -G) Minerals abundance (solid lines) and $MgO$ content (dotted lines) in the corresponding minerals. Panel 2-H) distribution of coesite. Local minerals abundance and compositions shown in panels 2-C) -H) are retrieved after performing thermodynamic computations at every spatial location with the program AlphaMELTS using the bulk oxides abundance exemplified in panel 2-B) for $MgO$. An animation file and complete data for all nine oxides are available following the instructions in the supplementary material. Time and distance in arbitrary units. Pressure and temperature are fixed at 40 kbar and $1200^oC$. The rest of the parameters for the model are defined in the main text.

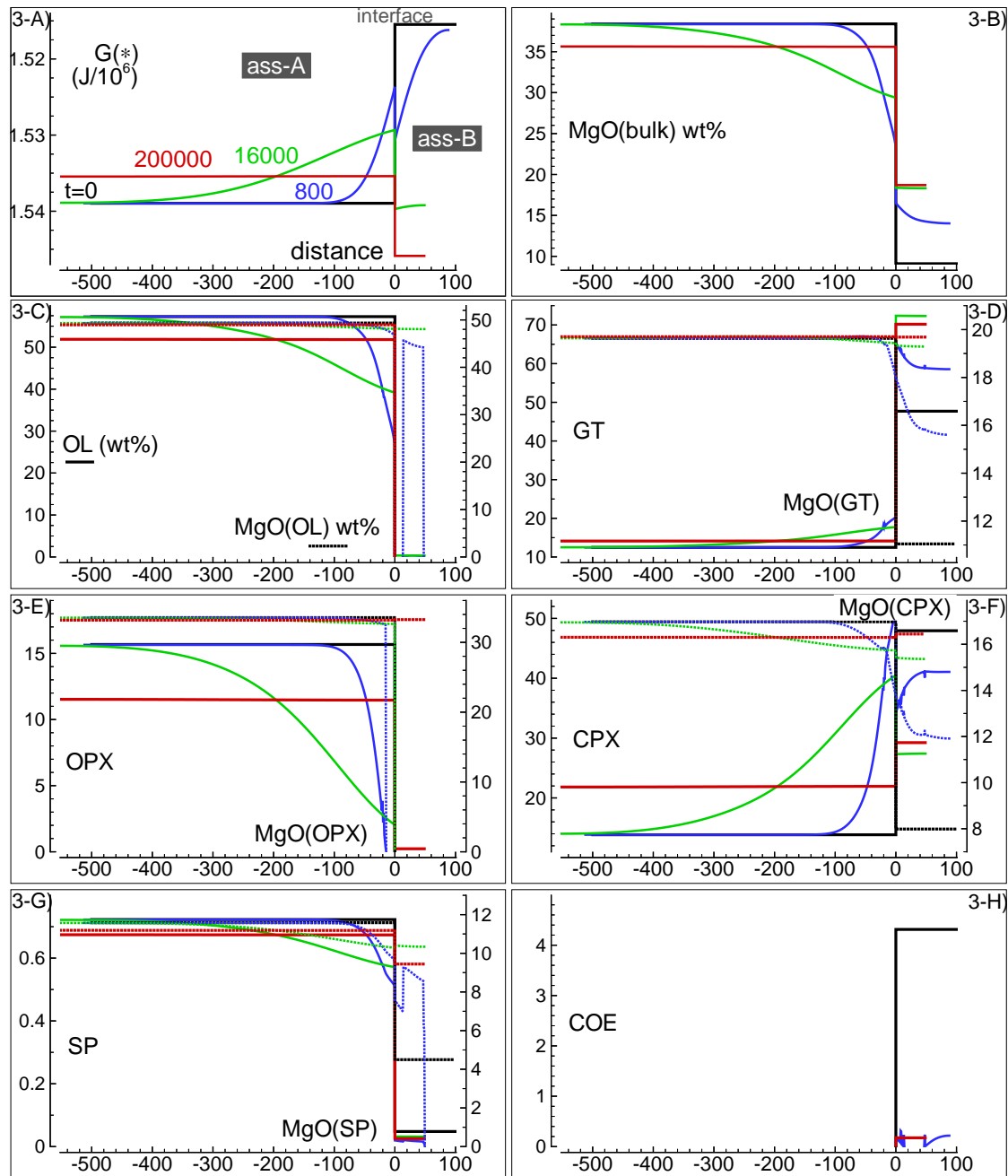

**Figure 3.** Solution of a 1-D model simulation. The initial proportion of the two assemblages is 5:1 ($f = 5$). The description of the panels follows the caption provided for figure 2.

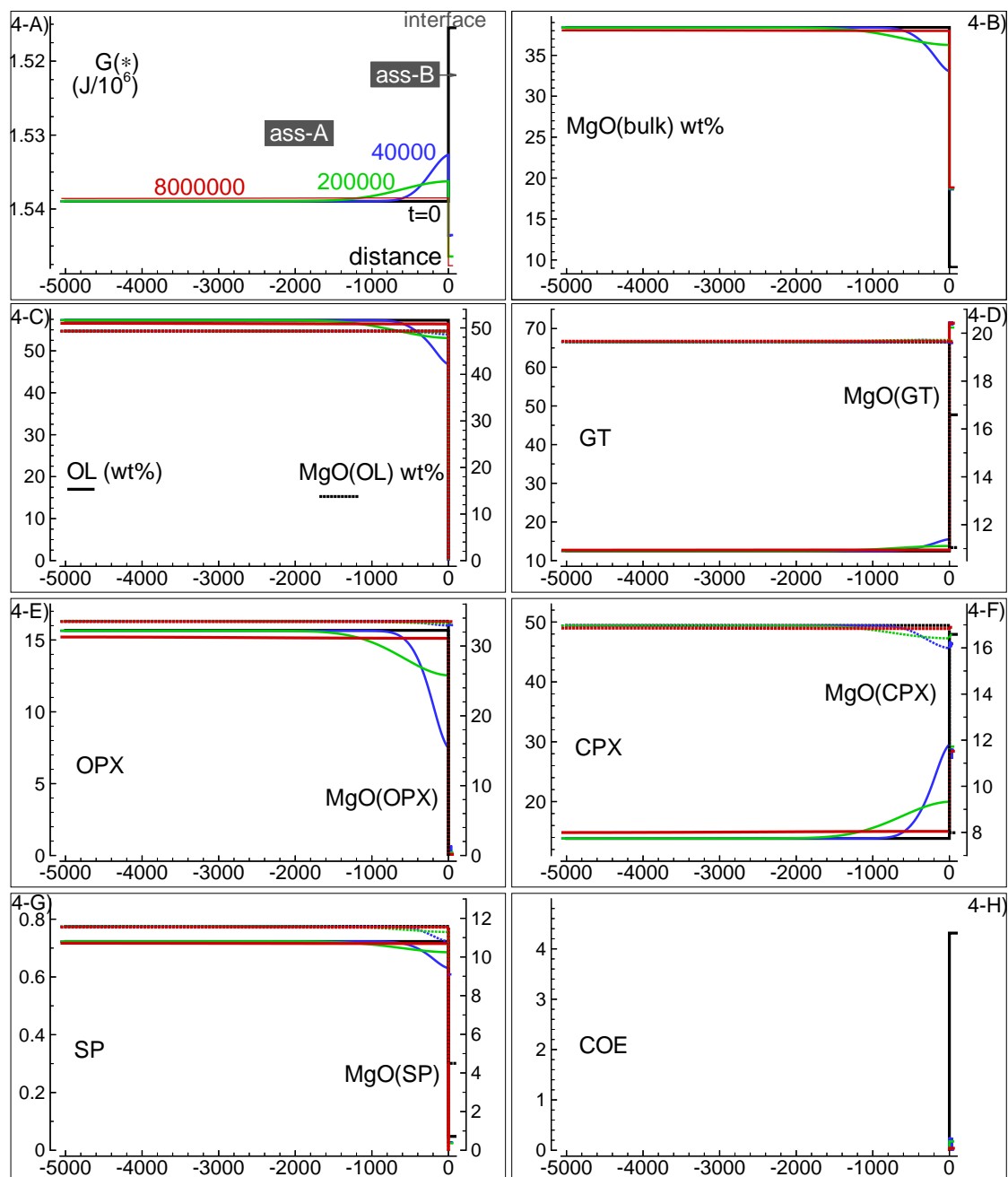

**Figure 4.** Solution for a 1-D model. The initial proportion of the two assemblages is 50:1 ($f = 50$). The description of the panels follows the caption provided for figure 2. Raw data file for all nine oxides can be retrieved online but no animation file is available for this simulation.

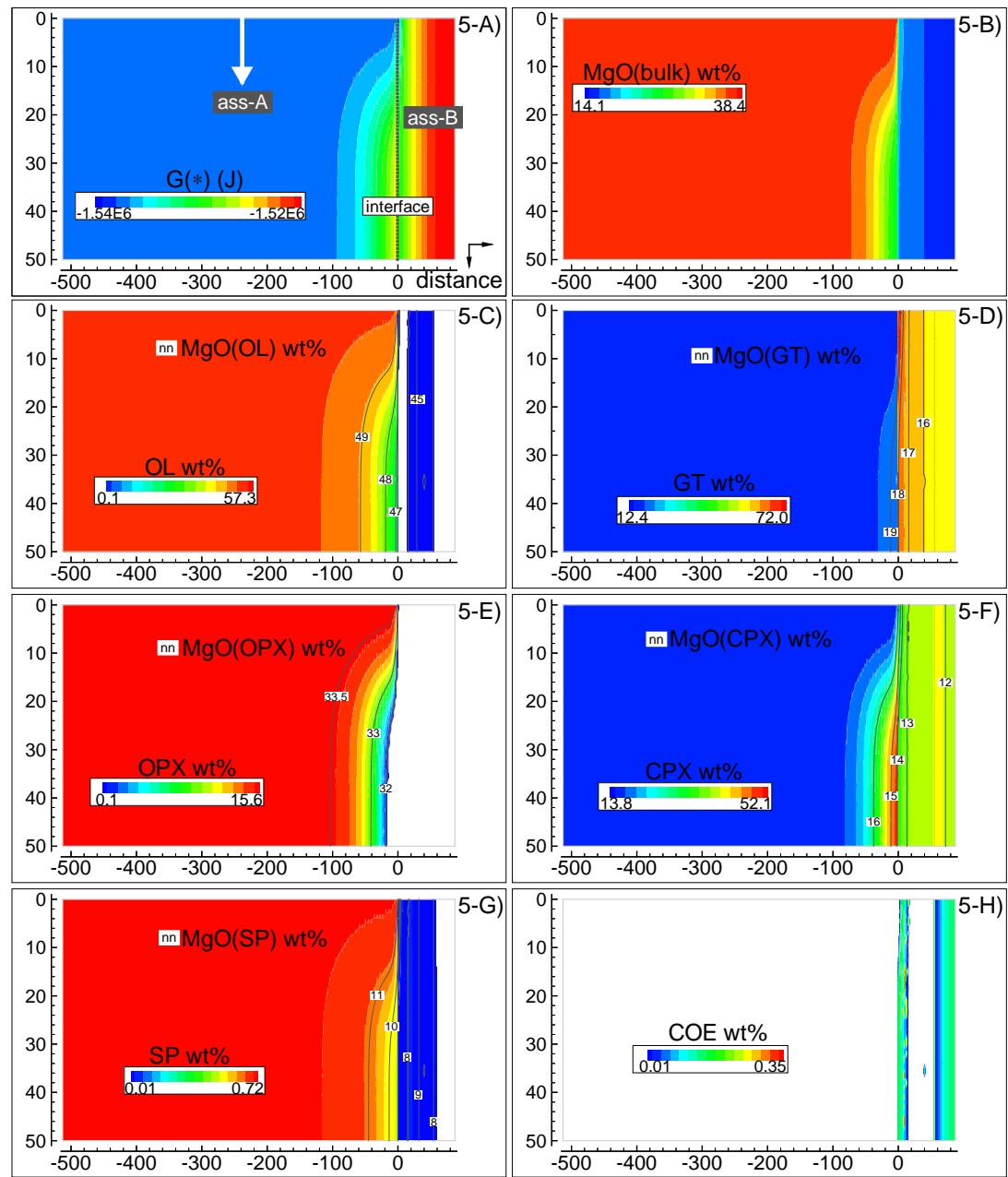

**Figure 5.** Solution of a 2-D model simulation at time 102400 (arbitrary units). The starting proportion of the two assemblages is 5:1 ($f = 5$). In the initial setup the 2 assemblages are separately in chemical equilibrium. At time 100000 a new assemblage $A$ enters from the top side with velocity 0.00625 (arbitrary units). The new assemblage is assumed to have been equilibrated but never previously in contact with assemblage $B$ (the composition of the new assemblage is the same of the assemblage in the initial setup). Panel 5-A) spatial variation of $G(*)$. Panel 5-B) local distribution of MgO in the bulk assemblage. Similar results are obtained for all the other oxides defining the bulk composition. An animation file and raw data for all nine oxides are available online following the instructions provided in the supplementary material. Panels 5-C) - G) local minerals distribution (color map) and few contour lines for the abundance of $MgO$ in the associate minerals. Panel 5-H) spatial distribution of coesite. Time and distance in arbitrary units. Pressure and temperature are fixed at 40 kbar and $1200^{o}$C. The rest of the parameters for the numerical model are defined in the main text.

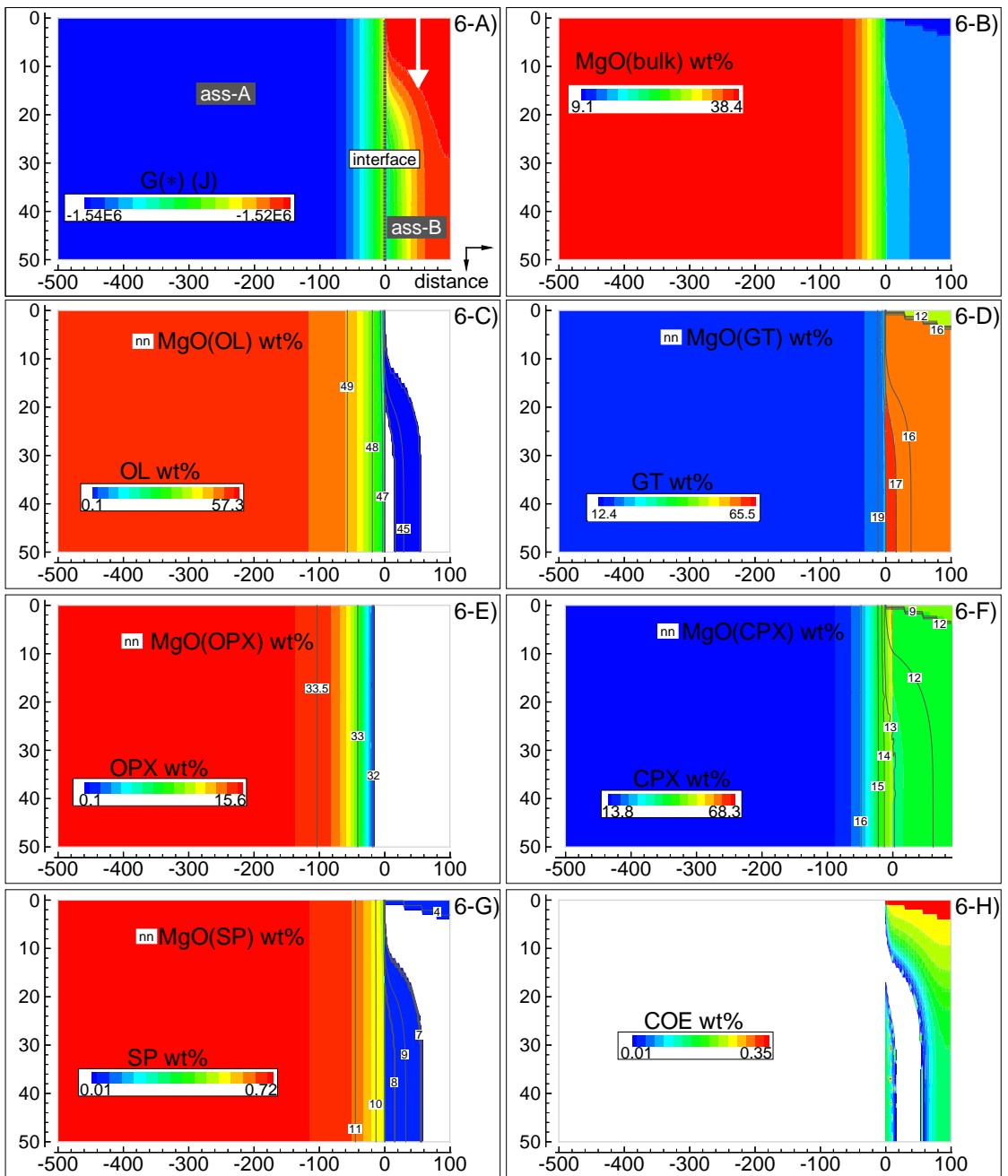

**Figure 6.** Solution of a 2-D model simulation at time 102400 (arbitrary units). The starting proportion of the two assemblages is 5:1 ($f = 5$). In this model it is assumed that at time 100000 a new assemblage $B$ enters from the top with velocity 0.00625 (arbitrary units). The description of the panels follows the caption of figure 5.

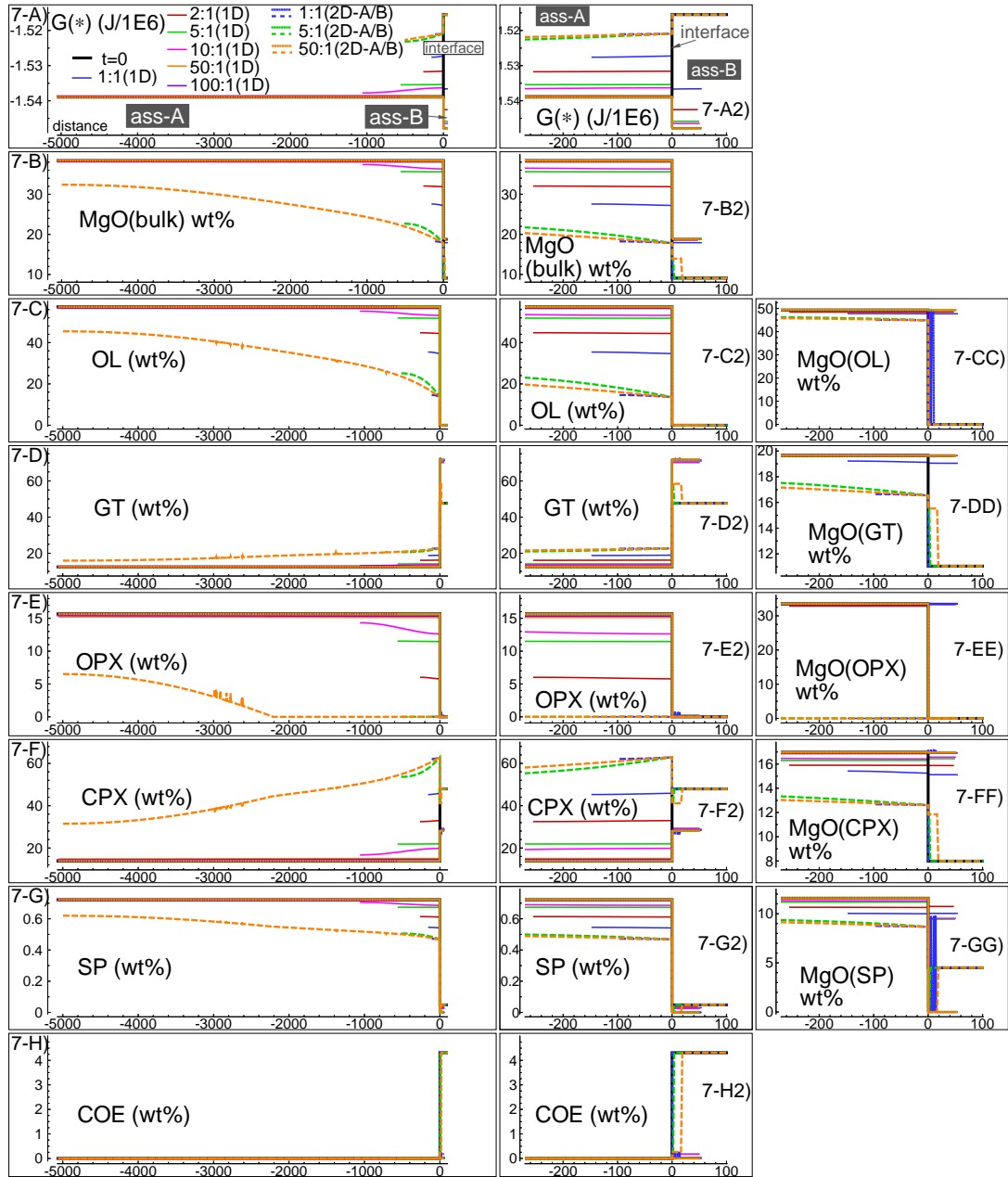

**Figure 7.** Summary of the results for all the 1-D and 2-D numerical models at conditions close to chemical equilibrium for the whole system. The models consider different initial proportions of the two assemblages. In addition for the 2-D models it is assumed that either assemblage $A$ or $B$ enters from the top side at time 100000 (arbitrary units) with velocity 0.00625 (arbitrary units). For the 2-D models the profiles represent an horizontal section at the vertical middle point ($D_y/2$). Panel 7-A) spatial variation of $G(*)$. For clarity, plot of the 2-D model with 50:1(B) is truncated at $x \sim 500$. Panel 7-A2) enlarged view of $G(*)$ near the interface. Panel 7-B) variation of bulk $MgO$ (wt%). Panel 7-B2) enlarged view of bulk $MgO$ near the interface. Panels 7-C) - G) spatial variation of minerals abundance. Panels 7-C2) - G2) minerals abundance zoomed near the interface. Panels 7-CC) - GG) $MgO$ content in the associated minerals near the interface. Panels 7-H) and 7-H2) distribution of coesite ($SiO_2$).

**Table 1.** List of minerals and mineral components relevant for this study with chemical formulas and abbreviations.

OLIVINE(Ol)

| | |
|---|---|
| fayalite(Fa) | $Fe_2^{2+}SiO_4$ |
| monticellite(Mtc) | $CaMgSiO_4$ |
| forsterite(Fo) | $Mg_2SiO_4$ |

GARNET(Gt)

| | |
|---|---|
| almandine(Alm) | $Fe_3^{2+}Al_2Si_3O_{12}$ |
| grossular(Grs) | $Ca_3Al_3Si_3O_{12}$ |
| pyrope(Prp) | $Mg_3Al_2Si_3O_{12}$ |

ORTHOPYROXENE(Opx) & CLINOPYROXENE(Cpx)

| | |
|---|---|
| diopside(Di) | $CaMgSi_2O_6$ |
| enstatite(en) | $Mg2Si_2O_6$ |
| hedenbergite(Hd) | $CaFe^{2+}Si_2O_6$ |
| alumino-buffonite(Al-Bff) | $CaTi_{0.5}Mg_{0.5}AlSiO_6$ |
| buffonite(Bff) | $CaTi_{0.5}Mg_{0.5}Fe^{3+}SiO_6$ |
| esseneite(Ess) | $CaFe^{3+}AlSiO_6$ |
| jadeite(Jd) | $NaAlSi_2O_6$ |

SPINEL(Sp)

| | |
|---|---|
| chromite(Chr) | $MgCr_2O_4$ |
| hercynite(Hc) | $Fe^{2+}Al_2O_4$ |
| magnetite(Mag) | $Fe^{2+}Fe_2^{3+}O_4$ |
| spinel(Spl) | $MgAl_2O_4$ |
| ulvospinel(Ulv) | $Fe_2^{2+}TiO_4$ |

COESITE(Coe)

| | |
|---|---|
| coesite(Coe) | $SiO_2$ |

**Table 2.** Set of independent reactions for the list of mineral components in table 1.

$$1.5\,\mathrm{Fa} + 1\,\mathrm{Prp} \quad \Leftrightarrow \quad 1.5\,\mathrm{Fo} + 1\,\mathrm{Alm} \tag{T-1}$$

$$1.5\,\mathrm{Fe}_2^{2+}\mathrm{SiO}_4 + 1\,\mathrm{Mg}_3\mathrm{Al}_2\mathrm{Si}_3\mathrm{O}_{12} \quad \Leftrightarrow \quad 1.5\,\mathrm{Mg}_2\mathrm{SiO}_4 + 1\,\mathrm{Fe}_3^{2+}\mathrm{Al}_2\mathrm{Si}_3\mathrm{O}_{12}$$

$$1\,\mathrm{Mtc} + 1\,\mathrm{OEn} \quad \Leftrightarrow \quad 1\,\mathrm{Fo} + 1\,\mathrm{ODi} \tag{T-2}$$

$$1\,\mathrm{CaMgSiO}_4 + 1\,\mathrm{Mg2Si}_2\mathrm{O}_6 \quad \Leftrightarrow \quad 1\,\mathrm{Mg}_2\mathrm{SiO}_4 + 1\,\mathrm{CaMgSi}_2\mathrm{O}_6$$

$$1\,\mathrm{Fa} + 0.5\,\mathrm{Fo} + 1\,\mathrm{OAlBff} + 1\,\mathrm{ODi} + 1\,\mathrm{OEss} \quad \Leftrightarrow \quad 2\,\mathrm{Mtc} + 1\,\mathrm{Alm} + 1\,\mathrm{OBff} \tag{T-3}$$

$$1\,\mathrm{Fe}_2^{2+}\mathrm{SiO}_4 + 0.5\,\mathrm{Mg}_2\mathrm{SiO}_4 + 1\,\mathrm{CaTi}_{0.5}\mathrm{Mg}_{0.5}\mathrm{AlSiO}_6 \quad + \quad 1\,\mathrm{CaMgSi}_2\mathrm{O}_6 + 1\,\mathrm{CaFe}^{3+}\mathrm{AlSiO}_6 \Leftrightarrow$$
$$2\,\mathrm{CaMgSiO}_4 + 1\,\mathrm{Fe}_3^{2+}\mathrm{Al}_2\mathrm{Si}_3\mathrm{O}_{12} + 1\,\mathrm{CaTi}_{0.5}\mathrm{Mg}_{0.5}\mathrm{Fe}^{3+}\mathrm{SiO}_6$$

$$0.5\,\mathrm{Fo} + 1\,\mathrm{OHd} \quad \Leftrightarrow \quad 0.5\,\mathrm{Fa} + 1\,\mathrm{ODi} \tag{T-4}$$

$$0.5\,\mathrm{Mg}_2^{2+}\mathrm{SiO}_4 + 1\,\mathrm{CaFe}^{2+}\mathrm{Si}_2\mathrm{O}_6 \quad \Leftrightarrow \quad 0.5\,\mathrm{Fe}_2^{2+}\mathrm{SiO}_4 + 1\,\mathrm{CaMgSi}_2\mathrm{O}_6$$

$$1\,\mathrm{CDi} \quad \Leftrightarrow \quad 1\,\mathrm{ODi} \tag{T-5}$$

$$1\,\mathrm{CaMgSi}_2\mathrm{O}_6 \quad \Leftrightarrow \quad 1\,\mathrm{CaMgSi}_2\mathrm{O}_6$$

$$1\,\mathrm{Mtc} + 1\,\mathrm{CEn} \quad \Leftrightarrow \quad 1\,\mathrm{Fo} + 1\,\mathrm{ODi} \tag{T-6}$$

$$1\,\mathrm{CaMgSiO}_4 + 1\,\mathrm{Mg2Si}_2\mathrm{O}_6 \quad \Leftrightarrow \quad 1\,\mathrm{Mg}_2\mathrm{SiO}_4 + 1\,\mathrm{CaMgSi}_2\mathrm{O}_6$$

$$0.5\,\mathrm{Fo} + 1\,\mathrm{CHd} \quad \Leftrightarrow \quad 0.5\,\mathrm{Fa} + 1\,\mathrm{ODi} \tag{T-7}$$

$$0.5\,\mathrm{Mg}_2\mathrm{SiO}_4 + 1\,\mathrm{CaFe}^{2+}\mathrm{Si}_2\mathrm{O}_6 \quad \Leftrightarrow \quad 0.5\,\mathrm{Fe}_2^{2+}\mathrm{SiO}_4 + 1\,\mathrm{CaMgSi}_2\mathrm{O}_6$$

$$1\,\mathrm{OAlBff} \quad \Leftrightarrow \quad 1\,\mathrm{CAlBff} \tag{T-8}$$

$$1\,\mathrm{CaTi}_{0.5}\mathrm{Mg}_{0.5}\mathrm{AlSiO}_6 \quad \Leftrightarrow \quad 1\,\mathrm{CaTi}_{0.5}\mathrm{Mg}_{0.5}\mathrm{AlSiO}_6$$

$$1\,\mathrm{OBff} \quad \Leftrightarrow \quad 1\,\mathrm{CBff} \tag{T-9}$$

$$1\,\mathrm{CaTi}_{0.5}\mathrm{Mg}_{0.5}\mathrm{Fe}^{3+}\mathrm{SiO}_6 \quad \Leftrightarrow \quad 1\,\mathrm{CaTi}_{0.5}\mathrm{Mg}_{0.5}\mathrm{Fe}^{3+}\mathrm{SiO}_6$$

$$1.5\,\mathrm{Fa} + 0.5\,\mathrm{Fo} + 1\,\mathrm{ODi} + 1\,\mathrm{OAlBff} + 1\,\mathrm{CEss} \quad \Leftrightarrow \quad 2\,\mathrm{Mtc} + 1\,\mathrm{Alm} + 1\,\mathrm{OBff} \tag{T-10}$$

$$1.5\,\mathrm{Fe}_2\mathrm{SiO}_4 + 0.5\,\mathrm{Mg}_2\mathrm{SiO}_4 + 1\,\mathrm{CaMgSi}_2\mathrm{O}_6 \quad + \quad 1\,\mathrm{CaTi}_{0.5}\mathrm{Mg}_{0.5}\mathrm{AlSiO}_6 + 1\,\mathrm{CaFe}^{3+}\mathrm{AlSiO}_6 \Leftrightarrow$$
$$2\,\mathrm{CaMgSiO}_4 + 1\,\mathrm{Fe}_3^{2+}\mathrm{Al}_2\mathrm{Si}_3\mathrm{O}_{12} + 1\,\mathrm{CaTi}_{0.5}\mathrm{Mg}_{0.5}\mathrm{Fe}^{3+}\mathrm{SiO}_6$$

$$1\,\mathrm{CJd} \quad \Leftrightarrow \quad 1\,\mathrm{OJd} \tag{T-11}$$

$$1\,\mathrm{NaAlSi}_2\mathrm{O}_6 \quad \Leftrightarrow \quad 1\,\mathrm{NaAlSi}_2\mathrm{O}_6$$

$$1.5\,\mathrm{Fa} + 1.5\,\mathrm{Fo} + 1\,\mathrm{Grs} \quad \Leftrightarrow \quad 3\,\mathrm{Mtc} + 1\,\mathrm{Alm} \tag{T-12}$$

$$1.5\,\mathrm{Fe}_2^{2+}\mathrm{SiO}_4 + 1.5\,\mathrm{Mg}_2^{2+}\mathrm{SiO}_4 + 1\,\mathrm{Ca}_3\mathrm{Al}_3\mathrm{Si}_3\mathrm{O}_{12} \quad \Leftrightarrow \quad 3\,\mathrm{CaMgSiO}_4 + 1\,\mathrm{Fe}_3^{2+}\mathrm{Al}_2\mathrm{Si}_3\mathrm{O}_{12}$$

$$1\,\mathrm{Fa} + 2\,\mathrm{ODi} + 1\,\mathrm{Hc} \quad \Leftrightarrow \quad 2\,\mathrm{Mtc} + 1\,\mathrm{Alm} \tag{T-13}$$

$$1\,\mathrm{Fe}_2^{2+}\mathrm{SiO}_4 + 2\,\mathrm{CaMgSi}_2\mathrm{O}_6 + 1\,\mathrm{Fe}^{2+}\mathrm{Al}_2\mathrm{O}_4 \quad \Leftrightarrow \quad 2\,\mathrm{CaMgSiO}_4 + 1\,\mathrm{Fe}_3^{2+}\mathrm{Al}_2\mathrm{Si}_3\mathrm{O}_{12}$$

$$1\,\mathrm{Fa} + 2\,\mathrm{OAlBff} + 2\,\mathrm{ODi} + 1\,\mathrm{Mag} \quad \Leftrightarrow \quad 2\,\mathrm{Mtc} + 1\,\mathrm{Alm} + 2\,\mathrm{OBff} \tag{T-14}$$

$$1\,\mathrm{Fe}_2^{2+}\mathrm{SiO}_4 + 2\,\mathrm{CaTi}_{0.5}\mathrm{Mg}_{0.5}\mathrm{AlSiO}_6 + 2\,\mathrm{CaMgSi}_2\mathrm{O}_6 \quad + \quad 1\,\mathrm{Fe}^{2+}\mathrm{Fe}_2^{3+}\mathrm{O}_4 \Leftrightarrow$$
$$2\,\mathrm{CaMgSiO}_4 + 1\,\mathrm{Fe}_3^{2+}\mathrm{Al}_2\mathrm{Si}_3\mathrm{O}_{12} + 2\,\mathrm{CaTi}_{0.5}\mathrm{Mg}_{0.5}\mathrm{Fe}^{3+}\mathrm{SiO}_6$$

$$1.5\,\mathrm{Fa} + 2\,\mathrm{ODi} + 1\,\mathrm{Spl} \quad \Leftrightarrow \quad 2\,\mathrm{Mtc} + 0.5\,\mathrm{Fo} + 1\,\mathrm{Alm} \tag{T-15}$$

$$1.5\,\mathrm{Fe}_2^{2+}\mathrm{SiO}_4 + 2\,\mathrm{CaMgSi}_2\mathrm{O}_6 + 1\,\mathrm{MgAl}_2\mathrm{O}_4 \quad \Leftrightarrow \quad 2\,\mathrm{CaMgSiO}_4 + 0.5\,\mathrm{Mg}_2\mathrm{SiO}_4 + 1\,\mathrm{Fe}_3^{2+}\mathrm{Al}_2\mathrm{Si}_3\mathrm{O}_{12}$$

$$2\,\mathrm{Mtc} + 1\,\mathrm{Alm} + 1\,\mathrm{Ulv} \quad \Leftrightarrow \quad 2\,\mathrm{Fa} + 0.5\,\mathrm{Fo} + 2\,\mathrm{OAlBff} \tag{T-16}$$

$$2\,\mathrm{CaMgSiO}_4 + 1\,\mathrm{Fe}_3^{2+}\mathrm{Al}_2\mathrm{Si}_3\mathrm{O}_{12} + 1\,\mathrm{Fe}_2^{2+}\mathrm{TiO}_4 \quad \Leftrightarrow \quad 2\,\mathrm{Fe}_2^{2+}\mathrm{SiO}_4 + 0.5\,\mathrm{Mg}_2^{2+}\mathrm{SiO}_4 + 2\,\mathrm{CaTi}_{0.5}\mathrm{Mg}_{0.5}\mathrm{AlSiO}_6$$

$$1\,\mathrm{Mtc} + 1\,\mathrm{Coe} \quad \Leftrightarrow \quad 1\,\mathrm{ODi} \tag{T-17}$$

$$1\,\mathrm{CaMgSiO}_4 + 1\,\mathrm{SiO}_2 \quad \Leftrightarrow \quad 1\,\mathrm{CaMgSi}_2\mathrm{O}_6$$

**Table 3.** Summary of the results of one chemical equilibration procedure. The columns ($A_0$) and ($B_0$) describe the initial bulk composition of the two sub-systems and the Gibbs free energy $G$ (joule) of the equilibrium assemblages separately. Following the AlphaMELTS input format, the bulk compositions are given in grams. The initial proportion of the whole system is f:1 (f=1) and the whole composition is reported in column ($W$). Columns ($A$) and ($B$) in the upper portion of the table present the results of the chemical equilibration in terms of oxides. Note that the sum of the oxides is not 100, which indicates a mass transfer between the two sub-systems. The columns in the lower part of the table shows the composition of the mineral components at equilibrium before the two sub-systems are put together (f×n($A_0$) and n($B_0$)) and after equilibration of the whole system (f×n(A) and n(B)). Change of moles (f×$\Delta$n(A), $\Delta$n(B) is also reported. The last column is the composition of the whole system ($W$) after equilibration.

**Table 3.**

| bulk comp. oxides(g) | $(A_0)$ | $(B_0)$ | $(W)=(f{\times}A_0+B_0)/(f+1)$ | (A) | (B) | |
|---|---|---|---|---|---|---|
| $SiO_2$ | 45.20 | 48.86 | 47.030 | 69.428 | 24.637 | |
| $TiO_2$ | 0.20 | 0.37 | 0.285 | 0.463 | 0.107 | |
| $Al_2O_3$ | 3.94 | 17.72 | 10.830 | 11.677 | 9.976 | |
| $Fe_2O_3$ | 0.20 | 0.84 | 0.520 | 0.852 | 0.188 | |
| $Cr_2O_3$ | 0.40 | 0.03 | 0.215 | 0.422 | 0.008 | |
| FeO | 8.10 | 7.61 | 7.855 | 11.116 | 4.600 | |
| MgO | 38.40 | 9.10 | 23.750 | 38.107 | 9.391 | |
| CaO | 3.15 | 12.50 | 7.825 | 11.565 | 4.089 | |
| $Na_2O$ | 0.41 | 2.97 | 1.690 | 2.736 | 0.643 | |
| sum | 100 | 100 | 100 | 146.367 | 53.639 | |
| G(J) | -1538956.549 | -1515471.201 | -1528524.097 | -2233778.043 | -823270.616 | |

| min. comp. | | | | mol | | | |
|---|---|---|---|---|---|---|---|
| f=1 | $f{\times}n(A_0)$ | $f{\times}\Delta n(A)$ | $f{\times}n(A)$ | $n(B_0)$ | $\Delta n(B)$ | $n(B)$ | $(f+1){\times}n(W)$ |
| Ol(Fa) | 0.0389399 | 0.0008090 | 0.0397489 | 0 | 0 | 0 | 0.0397490 |
| Ol(Mtc) | 0.0003421 | -0.0000555 | 0.0002867 | 0 | 0 | 0 | 0.0002867 |
| Ol(Fo) | 0.3504050 | -0.0726300 | 0.2777750 | 0 | 0 | 0 | 0.2777780 |
| Gt(Alm) | 0.0054726 | 0.0090575 | 0.0145301 | 0.0290995 | -0.0100502 | 0.0190492 | 0.0335803 |
| Gt(Grs) | 0.0035179 | 0.0039790 | 0.0074970 | 0.0347389 | -0.0248984 | 0.0098404 | 0.0173354 |
| Gt(Prp) | 0.0202554 | 0.0238298 | 0.0440852 | 0.0435766 | 0.0141234 | 0.0577001 | 0.1018422 |
| Opx(Di) | -0.0104230 | 0.0104500 | 0.0000000 | 0 | 0 | 0 | 0 |
| Opx(En) | 0.0700777 | -0.0700777 | 0.0000000 | 0 | 0 | 0 | 0 |
| Opx(Hd) | 0.0116778 | -0.0116778 | 0.0000000 | 0 | 0 | 0 | 0 |
| Opx(Al-Bff) | 0.0018136 | -0.0018136 | 0.0000000 | 0 | 0 | 0 | 0 |
| Opx(Bff) | -0.0003756 | 0.0003756 | 0.0000000 | 0 | 0 | 0 | 0 |
| Opx(Ess) | 0.0008425 | -0.0008425 | 0.0000000 | 0 | 0 | 0 | 0 |
| Opx(Jd) | 0.0021691 | -0.0021691 | 0.0000000 | 0 | 0 | 0 | 0 |
| Cpx(Di) | 0.0334109 | 0.1062036 | 0.1396146 | 0.0719139 | -0.0387234 | 0.0331905 | 0.1728462 |
| Cpx(En) | 0.0116014 | 0.0433811 | 0.0549825 | 0.0092274 | 0.0034382 | 0.0126656 | 0.0676615 |
| Cpx(Hd) | 0.0050948 | 0.0243636 | 0.0294585 | 0.0184485 | -0.0116133 | 0.0068352 | 0.0362970 |
| Cpx(Al-Bff) | 0.0017718 | 0.0024237 | 0.0041956 | 0.0178175 | -0.0167911 | 0.0010264 | 0.0052218 |
| Cpx(Bff) | 0.0016117 | 0.0056089 | 0.0072207 | -0.0085581 | 0.0101999 | 0.0016418 | 0.0088622 |
| Cpx(Ess) | -0.0001499 | 0.0029960 | 0.0028461 | 0.0190600 | -0.0183578 | 0.0007021 | 0.0035480 |
| Cpx(Jd) | 0.0110612 | 0.0772301 | 0.0882913 | 0.0958389 | -0.0750880 | 0.0207509 | 0.1090693 |
| Sp(Chr) | 0.0026319 | 0.0001425 | 0.0027745 | 0.0001974 | -0.0001432 | 0.0000542 | 0.0028287 |
| Sp(Hc) | -0.0014341 | 0.0002618 | -0.0011723 | -0.0000353 | 0.0000125 | -0.0000229 | -0.0011952 |
| Sp(Mag) | 0.0002881 | 0.0000133 | 0.0003014 | 0.0000092 | -0.0000033 | 0.0000059 | 0.0003073 |
| Sp(Spl) | 0.0020765 | -0.0001627 | 0.0019138 | 0.0000536 | -0.0000163 | 0.0000374 | 0.0019512 |
| Sp(Ulv) | 0.0000924 | -0.0000023 | 0.0000902 | 0.0000011 | 0.0000006 | 0.0000018 | 0.0000919 |
| Coe(Coe) | 0 | 0 | 0 | 0.0717690 | -0.0717690 | 0.0000000 | 0 |

**Table 4.** Normalized bulk composition $(A*)$ and $(B*)$ in the two sub-systems taken from the results of the model in table 3, $(A)$ and $(B)$. The lower part of the table shows the equilibrium mineral composition computed with the program AlphaMELTS for each sub-system separately.

**Table 4.**

| bulk comp. | $(A*)$ | $(B*)$ |
|---|---|---|
| oxides(g) | | |
| $SiO_2$ | 47.434 | 45.931 |
| $TiO_2$ | 0.316 | 0.199 |
| $Al_2O_3$ | 7.978 | 18.599 |
| $Fe_2O_3$ | 0.582 | 0.351 |
| $Cr_2O_3$ | 0.288 | 0.015 |
| FeO | 7.595 | 8.575 |
| MgO | 26.035 | 17.507 |
| CaO | 7.902 | 7.623 |
| $Na_2O$ | 1.869 | 1.199 |
| sum | 100 | 100 |
| G(J) | -1526157.990 | -1534831.832 |
| min. comp. | ——— mol ——— | |

| | n$(A*)$ | n$(B*)$ |
|---|---|---|
| Ol(Fa) | 0.0271722 | 0 |
| Ol(Mtc) | 0.0001954 | 0 |
| Ol(Fo) | 0.1897603 | 0 |
| Gt(Alm) | 0.0099353 | 0.0354870 |
| Gt(Grs) | 0.0051128 | 0.0184357 |
| Gt(Prp) | 0.0301249 | 0.1075543 |
| Opx(Di) | 0 | 0 |
| Opx(En) | 0 | 0 |
| Opx(Hd) | 0 | 0 |
| Opx(Al-Bff) | 0 | 0 |
| Opx(Bff) | 0 | 0 |
| Opx(Ess) | 0 | 0 |
| Opx(Jd) | 0 | 0 |
| Cpx(Di) | 0.0954926 | 0.0615373 |
| Cpx(En) | 0.0375875 | 0.0238162 |
| Cpx(Hd) | 0.0201308 | 0.0128313 |
| Cpx(Al-Bff) | 0.0028660 | 0.0018818 |
| Cpx(Bff) | 0.0049360 | 0.0030979 |
| Cpx(Ess) | 0.0019432 | 0.0012846 |
| Cpx(Jd) | 0.0603228 | 0.0386858 |
| Sp(Chr) | 0.0018958 | 0.0001013 |
| Sp(Hc) | -0.0008006 | -0.0000398 |
| Sp(Mag) | 0.0002063 | 0.0000046 |
| Sp(Spl) | 0.0013058 | 0.0000473 |
| Sp(Ulv) | 0.0000618 | 0.0000006 |
| Coe(Coe) | 0 | 0.0000130 |

**Table 5.** Summary of the results of a chemical equilibration procedure in which the initial composition of the two-sub-systems ($A_0$) and ($B_0$) is taken from the outcome of the previous model ($A*$ and $B*$ from table 4). The initial proportion of the whole system is f:1 (f=5). The description of the results follow the outline of the caption of table 3.

**Table 5.**

| bulk comp. | $(A_0)$ | $(B_0)$ | $(W)=(f\times A_0+B_0)/(f+1)$ | $(A)$ | $(B)$ |
|---|---|---|---|---|---|
| oxides(g) | | | | | |
| $SiO_2$ | 47.434 | 45.931 | 47.184 | 47.443 | 45.888 |
| $TiO_2$ | 0.316 | 0.199 | 0.297 | 0.317 | 0.200 |
| $Al_2O_3$ | 7.978 | 18.599 | 9.748 | 7.984 | 18.565 |
| $Fe_2O_3$ | 0.582 | 0.351 | 0.544 | 0.582 | 0.352 |
| $Cr_2O_3$ | 0.288 | 0.015 | 0.243 | 0.290 | 0.004 |
| FeO | 7.595 | 8.575 | 7.758 | 7.596 | 8.568 |
| MgO | 26.035 | 17.507 | 24.614 | 26.036 | 17.505 |
| CaO | 7.902 | 7.623 | 7.855 | 7.908 | 7.588 |
| $Na_2O$ | 1.869 | 1.199 | 1.757 | 1.869 | 1.199 |
| sum | 100 | 100 | 100 | 100.026 | 99.870 |
| G(J) | -1526157.990 | -1534831.832 | -1527602.900 | -1526543.811 | -1532898.134 |

| min. comp. | ——————————————————— mol ——————————————————— | | | | | | |
|---|---|---|---|---|---|---|---|
| f=5 | $f\times n(A_0)$ | $f\times \Delta n(A)$ | $f\times n(A)$ | $n(B_0)$ | $\Delta n(B)$ | $n(B)$ | $(f+1)\times n(W)$ |
| Ol(Fa) | 0.1358613 | -0.0000082 | 0.1358531 | 0 | 0 | 0 | 0.1358531 |
| Ol(Mtc) | 0.0009771 | 0.0000021 | 0.0009792 | 0 | 0 | 0 | 0.0009792 |
| Ol(Fo) | 0.9488016 | -0.0000419 | 0.9487596 | 0 | 0 | 0 | 0.9487596 |
| Gt(Alm) | 0.0496763 | 0.0000549 | 0.0497312 | 0.0354870 | -0.0000421 | 0.0354449 | 0.0851745 |
| Gt(Grs) | 0.0255638 | 0.0000723 | 0.0256361 | 0.0184357 | -0.0001625 | 0.0182731 | 0.0439087 |
| Gt(Prp) | 0.1506246 | 0.0001470 | 0.1507716 | 0.1075543 | -0.0001038 | 0.1074505 | 0.2582112 |
| Opx(Di) | 0 | 0 | 0 | 0 | 0 | 0 | 0 |
| Opx(En) | 0 | 0 | 0 | 0 | 0 | 0 | 0 |
| Opx(Hd) | 0 | 0 | 0 | 0 | 0 | 0 | 0 |
| Opx(Al-Bff) | 0 | 0 | 0 | 0 | 0 | 0 | 0 |
| Opx(Bff) | 0 | 0 | 0 | 0 | 0 | 0 | 0 |
| Opx(Ess) | 0 | 0 | 0 | 0 | 0 | 0 | 0 |
| Opx(Jd) | 0 | 0 | 0 | 0 | 0 | 0 | 0 |
| Cpx(Di) | 0.4774632 | 0.0004950 | 0.4779581 | 0.0615373 | -0.0002040 | 0.0613333 | 0.5392796 |
| Cpx(En) | 0.1879373 | -0.0003953 | 0.1875420 | 0.0238162 | 0.0002395 | 0.0240557 | 0.2115931 |
| Cpx(Hd) | 0.1006542 | -0.0000980 | 0.1005562 | 0.0128313 | 0.0000665 | 0.0128978 | 0.1134595 |
| Cpx(Al-Bff) | 0.0143300 | 0.0000554 | 0.0143854 | 0.0018818 | -0.0000249 | 0.0018568 | 0.0162418 |
| Cpx(Bff) | 0.0246801 | -0.0000725 | 0.0246076 | 0.0030979 | 0.0000431 | 0.0031409 | 0.0277448 |
| Cpx(Ess) | 0.0097160 | 0.0000429 | 0.0097589 | 0.0012846 | -0.0000210 | 0.0012637 | 0.0110218 |
| Cpx(Jd) | 0.3016142 | -0.0000509 | 0.3015633 | 0.0386858 | 0.0000065 | 0.0386923 | 0.3402993 |
| Sp(Chr) | 0.0094789 | 0.0000714 | 0.0095503 | 0.0001013 | -0.0000730 | 0.0000283 | 0.0095786 |
| Sp(Hc) | -0.0040030 | -0.0000297 | -0.0040327 | -0.0000398 | 0.0000279 | -0.0000120 | -0.0040447 |
| Sp(Mag) | 0.0010314 | 0.0000071 | 0.0010385 | 0.0000046 | -0.0000015 | 0.0000031 | 0.0010415 |
| Sp(Spl) | 0.0065290 | 0.0000523 | 0.0065813 | 0.0000473 | -0.0000278 | 0.0000195 | 0.0066009 |
| Sp(Ulv) | 0.0003088 | 0.0000019 | 0.0003107 | 0.0000006 | 0.0000003 | 0.0000009 | 0.0003116 |
| Coe(Coe) | 0 | 0 | 0 | 0.0000130 | -0.0000130 | 0.0000000 | 0 |

**Table 6.** Results from a chemical equilibration model with initial composition of the two sub-systems ($A_0$) and ($B_0$) analogous to the one presented in table 3. The only difference is that the initial proportion of the whole system is f:1 (f=5).

**Table 6.**

| bulk comp. oxides(g) | $(A_0)$ | $(B_0)$ | $(W)=(f \times A_0 + B_0)/(f+1)$ | $(A)$ | $(B)$ |
|---|---|---|---|---|---|
| $SiO_2$ | 45.20 | 48.86 | 45.810 | 50.424 | 22.744 |
| $TiO_2$ | 0.20 | 0.37 | 0.228 | 0.252 | 0.109 |
| $Al_2O_3$ | 3.94 | 17.72 | 6.237 | 5.619 | 9.322 |
| $Fe_2O_3$ | 0.20 | 0.84 | 0.307 | 0.340 | 0.141 |
| $Cr_2O_3$ | 0.40 | 0.03 | 0.338 | 0.404 | 0.008 |
| FeO | 8.10 | 7.61 | 8.018 | 8.837 | 3.928 |
| MgO | 38.40 | 9.10 | 33.516 | 38.364 | 9.279 |
| CaO | 3.15 | 12.50 | 4.708 | 4.910 | 3.700 |
| $Na_2O$ | 0.41 | 2.97 | 0.837 | 0.913 | 0.450 |
| sum | 100 | 100 | 100 | 110.064 | 49.683 |
| G(J) | -1538956.549 | -1515471.201 | -1535494.148 | -1689092.173 | -767503.430 |

| min. comp. | | | | — mol — | | | |
|---|---|---|---|---|---|---|---|
| f=5 | $f \times n(A_0)$ | $f \times \Delta n(A)$ | $f \times n(A)$ | $n(B_0)$ | $\Delta n(B)$ | $n(B)$ | $(f+1) \times n(W)$ |
| Ol(Fa) | 0.1946993 | 0.0044941 | 0.1991934 | 0 | 0 | 0 | 0.1991934 |
| Ol(Mtc) | 0.0017107 | -0.0001606 | 0.0015502 | 0 | 0 | 0 | 0.0015502 |
| Ol(Fo) | 1.7520250 | -0.0760450 | 1.6759800 | 0 | 0 | 0 | 1.6759784 |
| Gt(Alm) | 0.0273631 | 0.0094755 | 0.0368386 | 0.0290995 | -0.0127068 | 0.0163927 | 0.0532263 |
| Gt(Grs) | 0.0175897 | 0.0028033 | 0.0203930 | 0.0347389 | -0.0256505 | 0.0090884 | 0.0294782 |
| Gt(Prp) | 0.1012771 | 0.0293155 | 0.1305926 | 0.0435766 | 0.0144206 | 0.0579973 | 0.1886035 |
| Opx(Di) | -0.0521149 | 0.0111195 | -0.0409954 | 0 | 0 | 0 | -0.0409953 |
| Opx(En) | 0.3503883 | -0.0953800 | 0.2550083 | 0 | 0 | 0 | 0.2550059 |
| Opx(Hd) | 0.0583893 | -0.0133410 | 0.0450483 | 0 | 0 | 0 | 0.0450481 |
| Opx(Al-Bff) | 0.0090681 | -0.0028948 | 0.0061732 | 0 | 0 | 0 | 0.0061732 |
| Opx(Bff) | -0.0018783 | 0.0006532 | -0.0012251 | 0 | 0 | 0 | -0.0012250 |
| Opx(Ess) | 0.0042123 | -0.0011617 | 0.0030506 | 0 | 0 | 0 | 0.0030506 |
| Opx(Jd) | 0.0108455 | -0.0006791 | 0.0101664 | 0 | 0 | 0 | 0.0101663 |
| Cpx(Di) | 0.1670546 | 0.1163384 | 0.2833930 | 0.0719139 | -0.0415608 | 0.0303531 | 0.3137231 |
| Cpx(En) | 0.0580069 | 0.0600890 | 0.1180959 | 0.0092274 | 0.0030166 | 0.0122440 | 0.1303407 |
| Cpx(Hd) | 0.0254742 | 0.0267773 | 0.0522515 | 0.0184485 | -0.0129894 | 0.0054590 | 0.0577119 |
| Cpx(Al-Bff) | 0.0088591 | 0.0018465 | 0.0107056 | 0.0178175 | -0.0166661 | 0.0011514 | 0.0118564 |
| Cpx(Bff) | 0.0080586 | 0.0070392 | 0.0150978 | -0.0085581 | 0.0101264 | 0.0015683 | 0.0166634 |
| Cpx(Ess) | -0.0007496 | 0.0023225 | 0.0015728 | 0.0190600 | -0.0188731 | 0.0001868 | 0.0017596 |
| Cpx(Jd) | 0.0553062 | 0.0819615 | 0.1372677 | 0.0958389 | -0.0812992 | 0.0145396 | 0.1518248 |
| Sp(Chr) | 0.0131597 | 0.0001403 | 0.0133001 | 0.0001974 | -0.0001421 | 0.0000553 | 0.0133554 |
| Sp(Hc) | -0.0071704 | 0.0004160 | -0.0067544 | -0.0000353 | 0.0000073 | -0.0000281 | -0.0067824 |
| Sp(Mag) | 0.0014407 | -0.0000486 | 0.0013921 | 0.0000092 | -0.0000034 | 0.0000058 | 0.0013979 |
| Sp(Spl) | 0.0103828 | -0.0003637 | 0.0100191 | 0.0000536 | -0.0000120 | 0.0000416 | 0.0100607 |
| Sp(Ulv) | 0.0004622 | -0.0000514 | 0.0004108 | 0.0000011 | 0.0000006 | 0.0000017 | 0.0004125 |
| Coe(Coe) | 0 | 0 | 0 | 0.0717690 | -0.0717690 | 0.0000000 | 0 |

**Table 7.** Normalized bulk composition ($A*$) and ($B*$) of the two sub-systems taken from the results of the model in table 6. The lower part of the table shows the equilibrium mineral composition computed with the program AlphaMELTS for each sub-system separately.

**Table 7.**

| bulk comp. | $(A*)$ | $(B*)$ |
|---|---|---|
| oxides(g) | | |
| $SiO_2$ | 45.813 | 45.778 |
| $TiO_2$ | 0.229 | 0.219 |
| $Al_2O_3$ | 5.105 | 18.764 |
| $Fe_2O_3$ | 0.309 | 0.284 |
| $Cr_2O_3$ | 0.367 | 0.017 |
| FeO | 8.028 | 7.906 |
| MgO | 34.856 | 18.677 |
| CaO | 4.461 | 7.448 |
| $Na_2O$ | 0.830 | 0.907 |
| sum | 100 | 100 |
| G(J) | -1534650.844 | -1544800.044 |
| min. comp. | ——— mol ——— | |

| | n$(A*)$ | n$(B*)$ |
|---|---|---|
| Ol(Fa) | 0.0361962 | 0 |
| Ol(Mtc) | 0.0002817 | 0 |
| Ol(Fo) | 0.3045391 | 0 |
| Gt(Alm) | 0.0066953 | 0.0329652 |
| Gt(Grs) | 0.0037073 | 0.0183808 |
| Gt(Prp) | 0.0237244 | 0.1166920 |
| Opx(Di) | -0.0074620 | 0 |
| Opx(En) | 0.0464101 | 0 |
| Opx(Hd) | 0.0081985 | 0 |
| Opx(Al-Bff) | 0.0011239 | 0 |
| Opx(Bff) | -0.0002225 | 0 |
| Opx(Ess) | 0.0005551 | 0 |
| Opx(Jd) | 0.0018509 | 0 |
| Cpx(Di) | 0.0515058 | 0.0607473 |
| Cpx(En) | 0.0214049 | 0.0248836 |
| Cpx(Hd) | 0.0094773 | 0.0110775 |
| Cpx(Al-Bff) | 0.0019463 | 0.0023058 |
| Cpx(Bff) | 0.0027401 | 0.0031700 |
| Cpx(Ess) | 0.0002879 | 0.0003660 |
| Cpx(Jd) | 0.0249397 | 0.0292646 |
| Sp(Chr) | 0.0024168 | 0.0001111 |
| Sp(Hc) | -0.0012274 | -0.0000549 |
| Sp(Mag) | 0.0002532 | 0.0000099 |
| Sp(Spl) | 0.0018207 | 0.0000764 |
| Sp(Ulv) | 0.0000747 | 0.0000025 |
| Coe(Coe) | 0 | 0 |

*Supplementary Material:*

# Chemical Heterogeneities in the Mantle: Progress Towards a General Quantitative Description (revision II)

M. Tirone[1]

[1]No affiliation

**Correspondence:** M. Tirone (max.tirone@gmail.com)

## 1 Supplementary Data

This section describes the additional material available through an external data repository.

The link to access all the files is:

`https://figshare.com/s/9a97a1d047e783be8e54`

(Note: the private link will be revised and made public once the manuscript is accepted for publication.)

List of the available files:

- `TWOPD-G-KIN.DATA.ZIP`

- `TWOPD-G-KIN.MOVIE1.AVI`

- `TWOPD-G-KIN.MOVIE5.AVI`

- `2D-G-KIN.DATA.ZIP`

- `2D-G-KIN.MOVIE5A.AVI`

- `2D-G-KIN.MOVIE5B.AVI`

### 1.1 1-D Simulations

The zip file `TWOPD-G-KIN.DATA.ZIP` includes the data of three 1-D simulations assuming that the initial proportion of the two assemblages is 1:1, 5:1 and 50:1 ($f = 1, 5, 50$). The details of the models are discussed in the main text. For every simulation there are two data files: `TWOPD-G-KIN1.1.DAT` and `TWOPD-G-KIN2.1.DAT` for the case with 1:1 proportion,

`TWOPD-G-KIN1.5.DAT`, `TWOPD-G-KIN2.5.DAT` and `TWOPD-G-KIN1.50.DAT`, `TWOPD-G-KIN2.50.DAT` for the models with initial proportion 5:1 and 50:1, respectively. The data files are divided in blocks, each block of data refers to a

particular time step.

The first data file for each simulation (`TWOPD-G-KIN1.1.DAT.TWOPD-G-KIN1.5.DAT` and `TWOPD-G-KIN1.50.DAT`) includes in every block, distance, $G(*)$ (joules) and the grid step size for the two sub-systems. The number of grid points for sub-system $A$ and $B$ are 101 and 101 in the first simulation, 501 and 101 in the second simulation, 1001 and 101 in the third simulation. Time step is 4, 40 and 800 for the three simulations. Data are stored every 20, 20, 50 numerical time steps respectively. As discussed in the main text, time, distance and step size have arbitrary units.

The second data file of each simulation (`TWOPD-G-KIN2.1.DAT`, `TWOPD-G-KIN2.5.DAT` and `TWOPD-G-KIN2.50.DAT`) includes in every block, distance and abundance of nine oxides (grams or wt%) which describes the bulk composition at every grid point. The listed oxides are: $SiO_2$, $TiO_2$, $Al_2O_3$, $Fe_2O_3$, $Cr_2O_3$, $FeO$, $MgO$, $CaO$ and $Na_2O$.

Two 1-D animations `TWOPD-G-KIN.MOVIE1.AVI` and `TWOPD-G-KIN.MOVIE5.AVI` are based on the simulations with $f = 1$ and $f = 5$. The relative data are included in the zip file `TWOPD-G-KIN.DATA.ZIP`.

## 1.2  2-D Simulations

The results of two 2-D simulations are included in the zip file `2D-G-KIN.DATA.ZIP`. For both simulations the initial proportion of the two assemblages is set to 5:1. The interface between the two sub-systems is a vertical line. The first simulation assumes that assemblage $A$ becomes mobile downwards at time=1000000 (arbitrary units), while in the second simulation the dynamic assemblage is $B$. The velocity of the moving assemblages is set to 0.00625 (arbitrary units). New material entering from the top side has the same bulk composition of the initial assemblage. The composition is reported in the main text and in the data files here below. Output data are stored every 400 time steps and the simulation time step is 16 (arbitrary units). Each block of data defined by the label "ZONE" provides information related to a particular time step.

The first data file of each simulation (`2D-G-KIN1.5A.DAT` and `2D-G-KIN1.5B.DAT`) includes the distance x-direction, y-direction and $G(*)$. The number of grid points in the x-direction is 251 and 51 in sub-system $A$ and $B$, respectively (total initial distance is 500 and 100 in arbitrary units). The number of grid points in the y-direction is 51 (total distance is 50 in arbitrary units). A block of data is divided in sub-blocks. Each sub-block consists of $(251 + 51) \times 51$ data points. The first sub-block contains the x-coordinate defining the numerical grid, the second sub-block the y-coordinate and the third sub-block the $G(*)$ values at the corresponding grid points.

The second data file of each simulation (`2D-G-KIN2.5A.DAT` and `2D-G-KIN2.5B.DAT`) follows the same data structure, except that instead of $G(*)$, nine bulk oxides are listed in nine sub-blocks. The sequence of oxides is the same reported for the 1-D models.

The data in the zip file `2D-G-KIN.DATA.AVI` have been used to create two animations, `2D-G-KIN.MOVIE5A.AVI` and `2D-G-KIN.MOVIE5B.AVI`, both are available following the link to the external data repository.

## 2 Supplementary Table

The following table reports the initial bulk composition and the proportion factor $f$ of the two sub-systems for all the 43 cases considered in this study (see sections 2.1 and 2.2 in the main text).

**Table 1.** Initial bulk composition of the two assemblages and proportion factor $f$.

| bulk comp. | $(A_0)$ | $(B_0)$ | $(A_0)$ | $(B_0)$ | $(A_0)$ | $(B_0)$ | $(A_0)$ | $(B_0)$ | $(A_0)$ | $(B_0)$ |
|---|---|---|---|---|---|---|---|---|---|---|
| oxides (g or wt%) | #1(f=1) | | #2(f=1.2) | | #3(f=1.3) | | #4(f=1.6) | | #5(f=2) | |
| $SiO_2$ | 45.200 | 48.860 | 45.200 | 48.860 | 45.200 | 48.860 | 45.200 | 48.860 | 45.200 | 48.860 |
| $TiO_2$ | 0.200 | 0.370 | 0.200 | 0.370 | 0.200 | 0.370 | 0.200 | 0.370 | 0.200 | 0.370 |
| $Al_2O_3$ | 3.940 | 17.720 | 3.940 | 17.720 | 3.940 | 17.720 | 3.940 | 17.720 | 3.940 | 17.720 |
| $Fe_2O_3$ | 0.200 | 0.840 | 0.200 | 0.840 | 0.200 | 0.840 | 0.200 | 0.840 | 0.200 | 0.840 |
| $Cr_2O_3$ | 0.400 | 0.030 | 0.400 | 0.030 | 0.400 | 0.030 | 0.400 | 0.030 | 0.400 | 0.030 |
| FeO | 8.100 | 7.610 | 8.100 | 7.610 | 8.100 | 7.610 | 8.100 | 7.610 | 8.100 | 7.610 |
| MgO | 38.400 | 9.100 | 38.400 | 9.100 | 38.400 | 9.100 | 38.400 | 9.100 | 38.400 | 9.100 |
| CaO | 3.150 | 12.500 | 3.150 | 12.500 | 3.150 | 12.500 | 3.150 | 12.500 | 3.150 | 12.500 |
| $Na_2O$ | 0.410 | 2.970 | 0.410 | 2.970 | 0.410 | 2.970 | 0.410 | 2.970 | 0.410 | 2.970 |
| sum | 100 | 100 | 100 | 100 | 100 | 100 | 100 | 100 | 100 | 100 |
| | #6(f=5) | | #7(f=20) | | #8(f=100) | | #9(f=500) | | #10(f=1000) | |
| $SiO_2$ | 45.200 | 48.860 | 45.200 | 48.860 | 45.200 | 48.860 | 45.200 | 48.860 | 45.200 | 48.860 |
| $TiO_2$ | 0.200 | 0.370 | 0.200 | 0.370 | 0.200 | 0.370 | 0.200 | 0.370 | 0.200 | 0.370 |
| $Al_2O_3$ | 3.940 | 17.720 | 3.940 | 17.720 | 3.940 | 17.720 | 3.940 | 17.720 | 3.940 | 17.720 |
| $Fe_2O_3$ | 0.200 | 0.840 | 0.200 | 0.840 | 0.200 | 0.840 | 0.200 | 0.840 | 0.200 | 0.840 |
| $Cr_2O_3$ | 0.400 | 0.030 | 0.400 | 0.030 | 0.400 | 0.030 | 0.400 | 0.030 | 0.400 | 0.030 |
| FeO | 8.100 | 7.610 | 8.100 | 7.610 | 8.100 | 7.610 | 8.100 | 7.610 | 8.100 | 7.610 |
| MgO | 38.400 | 9.100 | 38.400 | 9.100 | 38.400 | 9.100 | 38.400 | 9.100 | 38.400 | 9.100 |
| CaO | 3.150 | 12.500 | 3.150 | 12.500 | 3.150 | 12.500 | 3.150 | 12.500 | 3.150 | 12.500 |
| $Na_2O$ | 0.410 | 2.970 | 0.410 | 2.970 | 0.410 | 2.970 | 0.410 | 2.970 | 0.410 | 2.970 |
| sum | 100 | 100 | 100 | 100 | 100 | 100 | 100 | 100 | 100 | 100 |
| | #11(f=1) | | #12(f=1.5) | | #13(f=2) | | #14(f=5) | | #15(f=20) | |
| $SiO_2$ | 47.434 | 48.860 | 47.434 | 48.860 | 47.434 | 48.860 | 47.434 | 48.860 | 47.434 | 48.860 |
| $TiO_2$ | 0.317 | 0.370 | 0.317 | 0.370 | 0.317 | 0.370 | 0.317 | 0.370 | 0.317 | 0.370 |
| $Al_2O_3$ | 7.978 | 17.720 | 7.978 | 17.720 | 7.978 | 17.720 | 7.978 | 17.720 | 7.978 | 17.720 |
| $Fe_2O_3$ | 0.582 | 0.840 | 0.582 | 0.840 | 0.582 | 0.840 | 0.582 | 0.840 | 0.582 | 0.840 |
| $Cr_2O_3$ | 0.288 | 0.030 | 0.288 | 0.030 | 0.288 | 0.030 | 0.288 | 0.030 | 0.288 | 0.030 |
| FeO | 7.595 | 7.610 | 7.595 | 7.610 | 7.595 | 7.610 | 7.595 | 7.610 | 7.595 | 7.610 |
| MgO | 26.035 | 9.100 | 26.035 | 9.100 | 26.035 | 9.100 | 26.035 | 9.100 | 26.035 | 9.100 |
| CaO | 7.902 | 12.500 | 7.902 | 12.500 | 7.902 | 12.500 | 7.902 | 12.500 | 7.902 | 12.500 |
| $Na_2O$ | 1.869 | 2.970 | 1.869 | 2.970 | 1.869 | 2.970 | 1.869 | 2.970 | 1.869 | 2.970 |
| sum | 100 | 100 | 100 | 100 | 100 | 100 | 100 | 100 | 100 | 100 |
| | #16(f=100) | | #17(f=500) | | #18(f=1.32) | | #19(f=2) | | #20(f=5) | |
| $SiO_2$ | 47.434 | 48.860 | 47.434 | 48.860 | 48.940 | 48.860 | 48.940 | 48.860 | 48.940 | 48.860 |
| $TiO_2$ | 0.317 | 0.370 | 0.317 | 0.370 | 0.393 | 0.370 | 0.393 | 0.370 | 0.393 | 0.370 |
| $Al_2O_3$ | 7.978 | 17.720 | 7.978 | 17.720 | 10.394 | 17.720 | 10.394 | 17.720 | 10.394 | 17.720 |
| $Fe_2O_3$ | 0.582 | 0.840 | 0.582 | 0.840 | 0.820 | 0.840 | 0.820 | 0.840 | 0.820 | 0.840 |
| $Cr_2O_3$ | 0.288 | 0.030 | 0.288 | 0.030 | 0.237 | 0.030 | 0.237 | 0.030 | 0.237 | 0.030 |
| FeO | 7.595 | 7.610 | 7.595 | 7.610 | 7.074 | 7.610 | 7.074 | 7.610 | 7.074 | 7.610 |
| MgO | 26.035 | 9.100 | 26.035 | 9.100 | 18.887 | 9.100 | 18.887 | 9.100 | 18.887 | 9.100 |
| CaO | 7.902 | 12.500 | 7.902 | 12.500 | 10.505 | 12.500 | 10.505 | 12.500 | 10.505 | 12.500 |
| $Na_2O$ | 1.869 | 2.970 | 1.869 | 2.970 | 2.751 | 2.970 | 2.751 | 2.970 | 2.751 | 2.970 |
| sum | 100 | 100 | 100 | 100 | 100 | 100 | 100 | 100 | 100 | 100 |

**Table 2.** (continue) Initial bulk composition of the two assemblages and proportion factor $f$.

| bulk comp. | $(A_0)$ | $(B_0)$ | $(A_0)$ | $(B_0)$ | $(A_0)$ | $(B_0)$ | $(A_0)$ | $(B_0)$ | $(A_0)$ | $(B_0)$ |
|---|---|---|---|---|---|---|---|---|---|---|
| oxides (g or wt%) | #21(f=20) | | #22(f=100) | | #23(f=500) | | #24(f=1) | | #25(f=10) | |
| $SiO_2$ | 48.940 | 48.860 | 48.940 | 48.860 | 48.940 | 48.860 | 49.619 | 48.860 | 49.619 | 48.860 |
| $TiO_2$ | 0.393 | 0.370 | 0.393 | 0.370 | 0.393 | 0.370 | 0.426 | 0.370 | 0.426 | 0.370 |
| $Al_2O_3$ | 10.394 | 17.720 | 10.394 | 17.720 | 10.394 | 17.720 | 11.372 | 17.720 | 11.372 | 17.720 |
| $Fe_2O_3$ | 0.820 | 0.840 | 0.820 | 0.840 | 0.820 | 0.840 | 0.918 | 0.840 | 0.918 | 0.840 |
| $Cr_2O_3$ | 0.237 | 0.030 | 0.237 | 0.030 | 0.237 | 0.030 | 0.219 | 0.030 | 0.219 | 0.030 |
| FeO | 7.074 | 7.610 | 7.074 | 7.610 | 7.074 | 7.610 | 6.745 | 7.610 | 6.745 | 7.610 |
| MgO | 18.887 | 9.100 | 18.887 | 9.100 | 18.887 | 9.100 | 16.074 | 9.100 | 16.074 | 9.100 |
| CaO | 10.505 | 12.500 | 10.505 | 12.500 | 10.505 | 12.500 | 11.518 | 12.500 | 11.518 | 12.500 |
| $Na_2O$ | 2.751 | 2.970 | 2.751 | 2.970 | 2.751 | 2.970 | 3.109 | 2.970 | 3.109 | 2.970 |
| sum | 100 | 100 | 100 | 100 | 100 | 100 | 100 | 100 | 100 | 100 |
| | #26(f=20) | | #27(f=100) | | #28(f=500) | | #29(f=1) | | #30(f=5) | |
| $SiO_2$ | 49.619 | 48.860 | 49.619 | 48.860 | 49.619 | 48.860 | 45.200 | 45.931 | 45.200 | 45.931 |
| $TiO_2$ | 0.426 | 0.370 | 0.426 | 0.370 | 0.426 | 0.370 | 0.200 | 0.199 | 0.200 | 0.199 |
| $Al_2O_3$ | 11.372 | 17.720 | 11.372 | 17.720 | 11.372 | 17.720 | 3.940 | 18.599 | 3.940 | 18.599 |
| $Fe_2O_3$ | 0.918 | 0.840 | 0.918 | 0.840 | 0.918 | 0.840 | 0.200 | 0.351 | 0.200 | 0.351 |
| $Cr_2O_3$ | 0.219 | 0.030 | 0.219 | 0.030 | 0.219 | 0.030 | 0.400 | 0.015 | 0.400 | 0.015 |
| FeO | 6.745 | 7.610 | 6.745 | 7.610 | 6.745 | 7.610 | 8.100 | 8.576 | 8.100 | 8.576 |
| MgO | 16.074 | 9.100 | 16.074 | 9.100 | 16.074 | 9.100 | 38.400 | 17.507 | 38.400 | 17.507 |
| CaO | 11.518 | 12.500 | 11.518 | 12.500 | 11.518 | 12.500 | 3.150 | 7.623 | 3.150 | 7.623 |
| $Na_2O$ | 3.109 | 2.970 | 3.109 | 2.970 | 3.109 | 2.970 | 0.410 | 1.199 | 0.410 | 1.199 |
| sum | 100 | 100 | 100 | 100 | 100 | 100 | 100 | 100 | 100 | 100 |
| | #31(f=20) | | #32(f=100) | | #33(f=500) | | #34(f=1) | | #35(f=5) | |
| $SiO_2$ | 45.200 | 45.931 | 45.200 | 45.931 | 45.200 | 45.931 | 45.200 | 45.914 | 45.200 | 45.914 |
| $TiO_2$ | 0.200 | 0.199 | 0.200 | 0.199 | 0.200 | 0.199 | 0.200 | 0.216 | 0.200 | 0.216 |
| $Al_2O_3$ | 3.940 | 18.599 | 3.940 | 18.599 | 3.940 | 18.599 | 3.940 | 18.582 | 3.940 | 18.582 |
| $Fe_2O_3$ | 0.200 | 0.351 | 0.200 | 0.351 | 0.200 | 0.351 | 0.200 | 0.296 | 0.200 | 0.296 |
| $Cr_2O_3$ | 0.400 | 0.015 | 0.400 | 0.015 | 0.400 | 0.015 | 0.400 | 0.005 | 0.400 | 0.005 |
| FeO | 8.100 | 8.576 | 8.100 | 8.576 | 8.100 | 8.576 | 8.100 | 8.015 | 8.100 | 8.015 |
| MgO | 38.400 | 17.507 | 38.400 | 17.507 | 38.400 | 17.507 | 38.400 | 18.551 | 38.400 | 18.551 |
| CaO | 3.150 | 7.623 | 3.150 | 7.623 | 3.150 | 7.623 | 3.150 | 7.459 | 3.150 | 7.459 |
| $Na_2O$ | 0.410 | 1.199 | 0.410 | 1.199 | 0.410 | 1.199 | 0.410 | 0.962 | 0.410 | 0.962 |
| sum | 100 | 100 | 100 | 100 | 100 | 100 | 100 | 100 | 100 | 100 |
| | #36(f=20) | | #37(f=100) | | #38(f=500) | | #39(f=1) | | #40(f=5) | |
| $SiO_2$ | 45.200 | 45.914 | 45.200 | 45.914 | 45.200 | 45.914 | 45.200 | 45.804 | 45.200 | 45.804 |
| $TiO_2$ | 0.200 | 0.216 | 0.200 | 0.216 | 0.200 | 0.216 | 0.200 | 0.281 | 0.200 | 0.281 |
| $Al_2O_3$ | 3.940 | 18.582 | 3.940 | 18.582 | 3.940 | 18.582 | 3.940 | 18.319 | 3.940 | 18.319 |
| $Fe_2O_3$ | 0.200 | 0.296 | 0.200 | 0.296 | 0.200 | 0.296 | 0.200 | 0.246 | 0.200 | 0.246 |
| $Cr_2O_3$ | 0.400 | 0.005 | 0.400 | 0.005 | 0.400 | 0.005 | 0.400 | 0.015 | 0.400 | 0.015 |
| FeO | 8.100 | 8.015 | 8.100 | 8.015 | 8.100 | 8.015 | 8.100 | 7.482 | 8.100 | 7.482 |
| MgO | 38.400 | 18.551 | 38.400 | 18.551 | 38.400 | 18.551 | 38.400 | 18.834 | 38.400 | 18.834 |
| CaO | 3.150 | 7.459 | 3.150 | 7.459 | 3.150 | 7.459 | 3.150 | 8.295 | 3.150 | 8.295 |
| $Na_2O$ | 0.410 | 0.962 | 0.410 | 0.962 | 0.410 | 0.962 | 0.410 | 0.723 | 0.410 | 0.723 |
| sum | 100 | 100 | 100 | 100 | 100 | 100 | 100 | 100 | 100 | 100 |

**Table 3.** (continue) Initial bulk composition of the two assemblages and proportion factor $f$.

| bulk comp. | $(A_0)$ | $(B_0)$ | $(A_0)$ | $(B_0)$ | $(A_0)$ | $(B_0)$ |
|---|---|---|---|---|---|---|
| oxides (g or wt%) | 41(f=20) | | 42(f=100) | | 43(f=500) | |
| $SiO_2$ | 45.200 | 45.804 | 45.200 | 45.804 | 45.200 | 45.804 |
| $TiO_2$ | 0.200 | 0.281 | 0.200 | 0.281 | 0.200 | 0.281 |
| $Al_2O_3$ | 3.940 | 18.319 | 3.940 | 18.319 | 3.940 | 18.319 |
| $Fe_2O_3$ | 0.200 | 0.246 | 0.200 | 0.246 | 0.200 | 0.246 |
| $Cr_2O_3$ | 0.400 | 0.015 | 0.400 | 0.015 | 0.400 | 0.015 |
| FeO | 8.100 | 7.482 | 8.100 | 7.482 | 8.100 | 7.482 |
| MgO | 38.400 | 18.834 | 38.400 | 18.834 | 38.400 | 18.834 |
| CaO | 3.150 | 8.295 | 3.150 | 8.295 | 3.150 | 8.295 |
| $Na_2O$ | 0.410 | 0.723 | 0.410 | 0.723 | 0.410 | 0.723 |
| sum | 100 | 100 | 100 | 100 | 100 | 100 |