# Peer review of "Chemical Heterogeneities in the Mantle: Progress Towards a General Quantitative Description (revision II)"

_Solid Earth, 2018_

## Referee Comment (RC1) · Anonymous Referee #1 · 15 Nov 2018

This manuscript reports AlphaMELTS computations of chemical equilibrium between peridotite + eclogite assemblage at 1200oC and 4 GPa. The author varies proportions of peridotite and links it to the change in bulk composition and mineral assemblage. The manuscript is concise and informative; however, there are a few major issues which I consider should be taken into account before publication.

Major comments 1) The manuscript lacks an application chapter and through that it also lacks relevant conclusions. I find that the entire "conclusions" chapter just repeats the introduction and choice of modelling but does not provide any real conclusions. There are a large number of experimental papers on peridotite/eclogite interaction reactions

(for instance, Yaxley and Green, 1998, Schweitz. Mineral. Petrol). There are also dozens of works on natural rocks, such as ophiolite complexes that contain lenses of eclogites and pyroxenites within peridotites (see works by G. Pearson). However, there is no attempt to compare the modelling results with the natural and experimental assemblage. In fact, the author states: "The results could be compared with existing data on melt products and residual solids observed in various geological settings to investigate indirectly, but from a quantitative perspective, the presence of chemical heterogeneities in the mantle". This should be done. 2) Currently, I find the abstract too complicated, unclear and somewhat vague. It does not provide 1) the scope of the study, 2) methods, 3) results and 4) conclusions. 3) English needs to be reviewed by a native-English speaker. In its current version there are too many mistakes and unclear sentences. 4) Some text is unclear. The following examples are not the only ones. Page 4. Lines 15-25. For instance, what is the difference between "bulk compositions of the two assemblages separately" and "weighted average of the bulk composition of the two assemblages in proportion 1:1". Tables: It al also unclear how "the composition of mineral components at equilibrium" in "the lower part of the table" can be negative. 5) Panels C-H in Figure 2 I find confusing and difficult to read. It is also not explained what causes spikes (zigzag distribution) in Cpx, Coe and Gt panels at 4000 time frame. 6) It will be good to explain in text what is shown in figures. How the proportions of minerals change and how these changes vary with the fractions of peridotite added.

Minor comments Page 3 Line 26. What is semi-general? Table 1. Formula of grossular is wrong. Al2. Enstatite, please, correct the subscript Table 2. Please, specify what are O and C in front of minerals? For instance, reaction (11), what is ODi? Likewise, equation (18), what is CBff and OBff?

---

## Author Comment (AC1) · 15 Dec 2018

I am truly grateful to the reviewer for taking the time to review the manuscript. My apologies for not posting a reply earlier, I was hoping to prepare a comprehensive response to both reviews and a submit a revised version of the manuscript all in once.

Given the difficulty to find a second reviewer, I decided to post a preliminary reply before revising and re-submitting the manuscript. Hopefully it will foster a further discussion.

==================================

[Figure]

On the comments of Referee#1:

Since the manuscript and the related work was done quite some time ago, I took the opportunity to read the abstract with fresh eyes and I have to agree with the reviewer that it may be helpful to make it more clear following the reviewer's feedback. This will be done in the revised version of the manuscript.
* * *
As for the application of the model to geological problems, this is the main objective of this study, however in my opinion at this stage it is premature. The reason is that we need experimental data for two purposes: 1- validate the model in the manuscript, 2- put a constraint on the extent of the equilibration process. Without such experiments and a reliable model to quantify the experimental results, it would be extremely difficult to apply the study of the manuscript to real problems. But this was clear to me since the beginning, in fact the manuscript is part of a larger project described in a proposal submitted recently (Nov/08/18) to the ERC-2019-SyG, ERC Synergy Program, "GEO-DIVE: Experimenting and Modeling Chemical GEO-DIVErsities in the Solid Earth". The project specifically describes the acquisition of the set of experimental data that are needed to constrain the process. These would be the first experiments of this kind applied to a coupled system peridotite-eclogite/dunite at high P,T in a piston cylinder and multi-anvil press (a preliminary experiment has been already conducted some time ago, I have the data). The only similar study that I found is the one by Martin et al., although the system and the objectives were different (Martin, A.M., Laporte, D., Koga, K.T., Kawamoto, T., Hammouda, T., 2011. Experimental Study of the Stability of a DolomiteCoesite Assemblage in Contact With Peridotite: Implications for Sediment-Mantle Interaction and Diamond Formation During Subduction. J. Petrol., 0, 1-27, doi:10.1093/petrology/egr066).

I was aware of the work by Yaxley and Green, the problem is that in their experiments, melt seems to be present in all cases, which makes the interpretation of the results

more difficult and perhaps not relevant for the problem described in the manuscript (and in the project).

The studies by G. Pearson are very interesting. These are the type of data that I am planning to use (if the project gets funded). In fact one of the applications described in the proposal is the study of the thermo petrological evolution of cratons. For brevity I won't get into details here but I'll be happy to share privately with the reviewer the ERC-2019-SyG proposal where more info have been included.

_______________-

I'll keep polishing the text and try to make it as clear as possible, thanks for the feedback! Just a few comments here below:

1- the meaning of "bulk compositions of the two assemblages separately" and "weighted average of the bulk composition of the two assemblages in proportion 1:1" is actually quite simple. The manuscript will be revised to make this point more understandable. Let's say that in assemblage A there is 50% of a certain oxide, and in assemblage B the abundance of the same oxide is 40%. If the two assemblages are put together to create a combined assemblage in proportion 1:1, the abundance of this oxide in the combined assemblage will be 45%. If the proportion is for example 2:1, the oxide in the combined assemblage is 46.667%. Just to complete the description, 45% (1:1 proportion) or 46.667% (2:1 proportion) does not mean that the oxide % at every spatial location in the combined assemblage will be 45% or 46.667%. These % numbers are only the *spatial average* for the combined system. The oxide on some location will be, say 42% or 44% or some other % (determined by the model).

2- the end-member components can be negative (eg. Sp(Hc) in table 3). This is how Mark Ghiorso decided to describe the solid mixture of certain mineral phases, like spinel. It is strange but not wrong, because what matters is that the sum of each oxide in the mineral is greater than zero. For example in Table 3, first column, component Sp(Hc) = -0.0014341. Now let's consider $Al_2O_3$, since $Al_2O_3$ is only present in

Sp(Hc, Fe2+Al2O4) and Sp(Spl, MgAl2O4) in the same proportion, as long as Sp(Spl) > Sp(Hc) the Al2O3 abundance in the mineral phase spinel will be greater than zero. This is indeed true in this case because Sp(Spl) = 0.0020765 > Sp(Hc).

3- in Figure 2, the zigzag patter of MgO in certain minerals is induced by a numerical error. The transport model determines the bulk oxide distribution in space and time, for example panel 2-B) shows the bulk Mgo, all good, no zigzag here. The next step is to use the bulk composition with the program AlphaMELTS to determine the equilibrium assemblage at every location (mineral abundance and composition). I believe that this is were the irregularities are created (numerically). I am not sure exactly about the reason. It could be that the abundance of certain mineral components is very small and not precisely determined by AlphaMELTS at the equilibrium point. I will investigate further and see if I can come up with a more precise answer, although I don't think I will be able to fix the problem, if it is truly related to AlphaMELTS.

———————————-

I'll be happy to provide further details or clarifications, there is still some time before the second review will materialize. In any case I promise to deliver a more prompt response!

---

## Referee Comment (RC2) · Anonymous Referee #2 · 17 Jan 2019

Review of Tirone, Chemical Heterogeneities in the Mantle. . .

I do not recommend publication of this paper.

This paper addresses an important problem in geophysics. Given a lithologically heterogeneous portion of the mantle consisting of two lithologies (A_0, B_0) with length scale (thickness) large compared with grain sizes, what is the nature and extent of chemical exchange between the two lithologies? The simplest answer, based on the inefficiency of chemical diffusion, is that the two lithologies do not chemically interact at all, maintaining their chemical compositions unchanged for geologically long periods of time, or at least until mantle convection has thinned them to such an extent that chem-

ical diffusion can operate. This end-member idealization has been widely explored in geophysics.

At the other extreme, assuming diffusion is sped up (perhaps by fluids) or the lithological thickness is sufficiently small for chemical diffusion to operate, the two lithologies may completely equilibrate, forming a new homogeneous lithology, W. This new homogeneous lithology is uniquely defined by equilibrium thermodynamics: given the compositions and relative amounts of $A\_0$ and $B\_0$, there is a unique equilibrium state W that results from minimization of the global Gibbs free energy.

The author explores something in between these two extremes. He allows for some, limited, reaction between $A\_0$ and $B\_0$, such that the two lithologies end up with altered, but still distinctive compositions, (A and B). This state is NOT uniquely given by equilibrium thermodynamics. It results from a CONSTRAINED minimization of the global Gibbs free energy. Depending on what one assumes for the nature of these constraints, one can achieve a whole host of non-equilibrium assemblages A,B.

Why would you impose constraints? It is unclear what the author's answer to this question might be. I might impose constraints because I might believe that a) chemical diffusion is limited but not zero and that b) some components might diffuse faster than others. Whether this is what the author has in mind or not is unclear.

We can examine the form of his constraints. These seem either to be thermodynamically unrealistic or ad hoc or both. Let's examine them in detail:

Aside from (trivial) mass conservation (Eq. 1), we come first to Eq. 2, which states that the chemical potentials of all components in A and B should be the same as they are in W. This makes no sense to me. How do the partially equilibrated lithologies A and B "know" about the chemical potentials in the true thermodynamic equilibrium state W? Consider an example:

$A\_0$: (1-f) moles of forsterite $B\_0$: f moles of fayalite W: olivine: $fo\_{(1-f)}fa\_f$ A: ? B: ?

The chemical potentials of fo and fa in A and B cannot be equal to those in W UNLESS, A=W and B=W. In other words, the constrained minimization in this case does provide an answer that differs from the unconstrained minimization. For any limited equilibration between A_0 and B_0, the chemical potential must differ.

The next paragraph (pg. 5 ll. 15-28) proposes another constraint based on inspection of phase assemblages in A_0, B_0, and W. In this case, one phase (olivine) is proposed to end up entirely in A with none in B, because this avoids the formation of a "new mineral" in B. But why should this be a constraint? I suppose there is some relationship here to minimizing mass exchange. But the formulation seems ad hoc. Such restrictions have consequences for other mineral exchange reactions and this seems to be the focus of the following paragraph (pg. 5 ll. 29-pg. 6 l. 6).

The next constraint is stated in Eq. 4. There is a potentially interesting notion behind this constraint that is not made explicit. It is in some way an attempt to minimize a generalized chemical driving force between the two lithologies, by minimizing the difference in Gibbs free energy between them. This is intriguing, although why one should focus on minimizing the chemical driving force, rather than minimizing the mass exchange is not clear to me. It is worth pointing out that while the left-hand side of Eq. 4 can be minimized, the equation as written cannot be satisfied, even in simple systems, as the author showed in his previous study (2016).

The last constraint, Eq. 5, seems to derive from the arguments already presented (ca. Eq. 5). Not sure why an additional constraint is needed here.

The author then goes on to consider time-dependent problems, for which Eq. 6 seems to be the basis. This is obviously a diffusion equation and S clearly plays the role of a diffusivity. So why insist that S is dimensionless? Some more physical motivation is needed here. By the way, I do not understand the "*" notation. Is the composition $A*=A$? If so, wouldn't $G*(A)$ be better notation then $G(A*)$ for the normalized Gibbs free energy? Also, what is the normalized Gibbs free energy? I do not see it defined.

A final comment. The paper is completely untethered to observations, either from experiment or the field. Rocks consisting of finely interlayer eclogite/peridotite are well studied. Surely there is some opportunity for comparison? I note that the previous paper (Tirone et al., 2016) made some comparison with the experiments of Milke et al. (2007), although that study emphasized the importance of the relative diffusion rates of different chemical components, which Tirone does not seem to consider, or perhaps only in an indirect way via Eq. 6.

---

## Editor Comment (EC1) · Heap (Editor) · 18 Jan 2019

Dear Dr. Tirone,

As you can see, I've now received two reviews of your manuscript. These reviewers recommend "major revision" and "reject", respectively. Both reviewers, in my opinion, raise legitimate concerns. If you are keen to revise your paper, please now prepare a detailed point-by-point rebuttal letter, paying particular attention to the comments of Reviewer #2, and a revised manuscript. Please note that I consider that your manuscript will require at least another round of review before it can be reconsidered for publication in Solid Earth, and that the invitation to revise is not a guarantee that the paper will

eventually be accepted for publication. If you decline to revise your manuscript, it will remain online, together with the discussion comments, in Solid Earth Discussions.

Thanks for submitting your work to Solid Earth,

Mike (Topical Editor of Solid Earth)
* * *

---

## Author Comment (AC2) · 18 Jan 2019

Thanks for the hard work put to handle the manuscript.

I am very excited about the second review, it makes almost the perfect example for why an open review process should be the norm for all journals. I detailed reply to review#2 is on its way, coming up in few days.

The manuscript will be revised accordingly and resubmitted, it will take a bit more than few days though, but hopefully in the meantime there will be more discussion posted online, I am all in, would that be possible?

[Figure]

Another round of review sounds perfect to me!

My apologies for extending the editorial burden for this manuscript, but I strongly believe that it is worth it.

max

—————————————————————

---

## Author Comment (AC3) · 22 Jan 2019

I truly appreciate the effort and the time that the reviewer(#2) has put into this manuscript.

It is also a fantastic opportunity to be able to respond in a public forum, thanks to Solid Earth and the open review policy of the journal.

General comments:

______________________________________-

It is true that the physics of mechanical mixing in the mantle has been addressed for

many years now in the "end-member idealization" (as referred by the reviewer) where two lithologies are forever impermeable to mass exchange (the early studies by Kellog, for example, are very interesting). However we know that some chemical exchange always takes place on a certain spatial scale (this end-member case is strictly correct only for contact-time = 0), but no effort has been put to quantify the approximation introduced by the end-member case, why?

The idea that chemical equilibration is too slow to be effective has been permeating the solid Earth community for over forty years. Unfortunately there are no modern published evidences that support such conclusion nor quantify it. The concept of chemical diffusion is also often used quite freely either to dismiss any further investigation or to make general assumptions. But which diffusion mechanism? what chemical elements? which minerals are affected and how much? what other processes are involved and how? The "simple answer" (quoting the reviewer) that diffusion is too slow for the problem in hand is really too simple.

The field of chemical kinetics is also much more than "diffusion", there are several excellent books describing the complexity of chemical processes, for example "Chemical Kinetics of Solids" by H. Schmalzried. Perhaps from a geophysics perspective an analogy can be made between the chemical equilibration processes and the processes in the deformation creep. Even for a monomineralic policrystalline assemblage elemental volume diffusion (e.g. Si) does not entirely describe the various creep models. And when a two minerals assemblage is considered, the deformation can assume a large range of values depending on several factors (a couple of well known references are worth to mention: Karato's book, Deformation of Earth materials; and Takeda, Flow in rocks modelled as multiphase continua: application to polymineralic rocks. J. Struct. Geol., 1998). What about deformation for a 4-5 minerals assemblage?

Back to the chemical equilibration problem. The early work by Hofmann, which by the way was great for that time (1978), has been already discussed in the manuscript. I am not aware of further studies that substantiate and quantify experimentally or theoretically the extent and the nature of the chemical equilibration between two multiphase multicomponent assemblages (peridotite-eclogite or similar). How far the eclogite and peridotite equilibrate at 1300C, 50 kbar after 1 Myr? and after 1Byr? what about 80 kbar? what is the composition/mineralogy? Any reference would be greatly appreciated.

Comments on specific issues:

————————————————-

Perhaps there is a misunderstanding on what is the other end-member extreme that the reviewer refers as a "new homogeneous lithology". From a chemical perspective it does not happen. When the two sub-systems (lithologies) A and B are in chemical equilibrium they will not homogenize. They will preserve distinctive chemical and mineralogical features but different from those in the initial state (before equilibration). For a fixed proportion of A0 and B0, this is the other end-member state, it is not an intermediate state. Perhaps the easiest way to see this is by considering the reaction between qz and periclase to form forsterite(fo) and enstatite(en):

$MgO + n\ SiO_2 \rightarrow (1-n)\ Mg_2SiO_4 + (2n-1)\ MgSiO_3$

As shown for example by Gardes et al. (CMP, 161, 1-12, 2011), in complete equilibrium fo and en will not mix to form an homogeneous bimineralic single layer but instead there will be two separate layers (A and B), one made of polycristalline pure fo and the other by polycristalline pure en.

————————————

The reviewer called the computation described in the manuscript a constrained minimization of the Gibbs free energy, I don't think it is correct, although I am not quite sure. Minimization of the Gibbs energy is applied to the whole system (W), without any constraint beside the usual mass balance. The standard procedure requires as input quantities P,T and bulk composition (computed from A0 + B0 in a predefined proportion). The output is the mineral phases composition and abundance for the whole system W. Clearly there are no info regarding the two sub-systems A,B. Using the analogy of the example with qz and periclase, the G minimization applied to the whole system (qz+MgO) will determine that en and fo will be formed in equilibrium and it will define their abundance. For completeness, for this simple example mass balance would have been sufficient to determine the abundance, but in any case, the key point is that no indication is provided regarding the spatial distribution of fo and en.

In the manuscript the constraints are applied in a subsequent step. The answer to the question on why do we need to impose chemical constraints, is quite simply actually, because we want to know what is the composition and mineralogy of A and B after they have been put together and they reached chemical equilibrium. Translating the task in the simple example above, the question would concern the spatial distribution of fo and en. As mentioned by the reviewer (I think), something similar has been done for an analogous simple system in the previous study (Tirone, 2015), that is the reaction qz + fo -> en. But we want to push things forward, right? The main problem here is that there are no experimental data to validate the procedure that defines A and B for much more complex systems (described by 9 oxides), like those considered in this manuscript.

Now looking back at the way the constraints have been presented in the manuscript I am realizing that I could have done a better job. In particular making first a list of the constraints that are not based on any assumption, those are eqs. 1-4: eq.1) mass conservation, eq.2) equality of the chemical potentials for the same components (more on this later), relations similar to eq.3) to constrain the extent of certain reactions and eq.4) conservation of the total G. The last equation, eq.4, simply imposes that the sum of the Gibbs energy on the two sides (A and B) must be equal to the Gibbs energy of the whole system (W)

Unfortunately these relations are not sufficient to determine a unique set of compositions in A and B after equilibration, hence certain assumptions need to be made. For example olivine is initially present only in A0 but not in B0. After equilibration, in the

whole assemblage a certain amount of olivine must be present (see last column of the tables). Where would this olivine be located? As a first approximation the assumption is that the various reactions will change the composition of the preexisting olivine in A while no new olivine will be formed on the B side. As mentioned previously and also in the manuscript, only new experimental data can validate this type of assumptions.
* * *
if I understand correctly, the reviewer question is the following: why in phases described by the same components the chemical potential of the components must be equal when the system is in chemical equilibrium? This is one of those fundamental principles discussed in many books on chemical thermodynamics, my favorite are the classics (e.g Chemical Thermodynamics, Prigogine & Defay; The Principles of Chemical Equilibrium, Denbigh). The following simple explanation is based on material I used for teaching a while ago. Let's say we have two systems, A,B each made only of olivine but with different compositions Ol(A) made of two components, Fo(A), Fa(A) and Ol(B) made of Fo(B), Fa(B). When they are put together to form a larger system (A+B), two and only two independent reactions can be used to describe the chemical equilibrium (see for example the book: Chemical Reaction Equilibrium Analysis by Smith & Missen). The choice of the two reactions is arbitrary (for other combinations the conclusion will be the same), let's pick the simplest reactions:

(1) Fo(A) <-> Fo(B)

(2) Fa(A) <-> Fa(B)

based on the stoichiometry of the reactions it is obvious the changes of the moles "dn" are related:

dn_Fo(A) = -dn_Fo(B)

and

dn_Fa(A) = -dn_Fa(B)

Let's now recall the well known differential relation for G:

dG = -S dT + V dP + sum_i mu_i dn_i

where the sum is over the moles of the "i" independent components describing the system, and mu_i is the chemical potential of these components.

Therefore for the large system (A+B) at constant T, and P we have:

dG_(A+B) = mu_Fo(A) dn_Fo(A) - mu_Fo(B) dn_Fo(A) + mu_Fa(A) dn_Fa(A) - mu_Fa(B) dn_Fa(A)

The minimum of the Gibbs energy defining the equilibrium condition is given by two relations:

dG_(A+B)/dn_Fo(A) = 0 = mu_Fo(A) - mu_Fo(B)

and

dG_(A+B)/dn_Fa(A) = 0 = mu_Fa(A) - mu_Fa(B)

It is trivial to show that mu_Fo(A) - mu_Fo(B) = 0 can be reduced to n_Fo(A) = n_Fo(B) (and similarly for Fa) when olivine is described by an ideal mixing model. The equal number of moles condition was valid in the previous study, but not in this manuscript because this study relies on the thermo database by Ghiorso (and Berman).

————————————————

The idea of using the extensive function G to describe the chemical changes in the 2 sub-systems over time and space is a mean to simplify a problem that otherwise becomes intractable for complex systems. However the choice is not a complete abstraction, it is based on the consideration that the mass exchange is not governed by the compositional gradient but by the differences in the chemical potential of the various components in the various phases (the book by Denbigh has a nice discussion about this topic).

Quoting from <Deformation Mechanism Maps> (Frost & Ashby, 1982, Ch.1): "Both the equations in the following sections, and the maps constructed from them, must be regarded as a first approximation only. The maps are no better (and no worse) than the equations and data used to construct them."
* * *
Regarding the dynamic models: the reviewer is correct that the parameter "S" in eq.6 and eq.9 serves the same purpose of the thermal or chemical diffusivity terms in the heat conduction eq and Fick's laws. The parameter S is dimensionless because the problem is dimensionless, no units are used for spatial and temporal variations. Once again, we don't have the experimental data to put real dimensions to these numbers.
* * *
About G(A*): the definition of A* and B* was introduced on pag. 7 line 7, as the mineralogical composition and abundance of the assemblages in A and B normalized to oxide bulk abundance = 100. But I agree that it should have been presented more clearly in the main text, including the relation with G(A*), G(B*) and the more general G(*), S(*).
* * *
As I mentioned also in the reply to review#1 the comparison with field observations is premature. There are no experimental data to validate the model and to constrain the extent of chemical equilibration. That's also why the title of this manuscript "Progress towards...". It is just a mere step forward not the end of the story.

If (huge IF) a project related to this problem will get funded (ERC-2019-SyG, ERC Synergy Program, "GEO-DIVE: Experimenting and Modeling Chemical GEO-DIVErsities in the Solid Earth"), we will finally acquire the first experimental data for the peridotite-eclogite equilibration and we will be able to apply the model to some real geological problems. Those mentioned in the ERC proposal are the evolution of cratons and

thermochemical plumes.

Incidentally it also seems that I am the only one who has the data of the the first successful experiment of this kind on a peridotite-dunite couple, no melt or fluid involved (many thanks to Stephan Buhre at the IG in Mainz who carried out the exp). The preliminary (unpublished) results have been included in the ERC funding proposal.

————————————

Final Remark: I am strongly convinced that the manuscript is worth publishing since progress on such important topic in solid Earth has been stagnant for way too long.

I will make the necessary changes to the manuscript that, if the editor agrees, will be resubmitted.

In the meantime, assuming the online interacting forum will stay open, it would be really helpful to receive comments, requests for clarification, questions etc. (in alternative I'll gladly respond by email).

max

p.s. One may wonder why not doing the experiments first and then develop all the models and applications later. In a perfect world it would make sense. Unfortunately I already tried to get funds for the experimental work, twice, both attempts failed. The aforementioned ERC project is my last shot.

—————————————————————

---

## Author Response (AR1)

Massimiliano Tirone
email: max.tirone@gmail.com

January 29, 2019

Executive Editor,
Solid Earth

This letter complements the resubmission of the manuscript "Chemical Heterogeneities in the Mantle: Progress Towards a General Quantitative Description" by M. Tirone.

Taking into consideration the comments of the two reviewers, the manuscript has been extensively revised to make it more clear and understandable. The text has been also polished quite a bit.

Perhaps the major change involved the presentation of the constraints to determine the equilibrium composition in the two sub-systems once the two lithologies are put together and they have reached thermodynamic equilibrium as a whole. In particular a clear distinction is made between the relations that are generally valid and the constraints based on certain assumptions.

Both reviews asked for some comparison of the model results with field observations or experimental data. Experimental data are not available for the type of system considered in this study. I agree that field observations are necessary to validate the model, however as mentioned in the replies to the reviewers, it would not make much sense at this point to relate the results with real data since some experiments are absolutely necessary before moving forward. I think I also made quite clear in the replies why such experiments were not performed earlier.

The most difficult part of the revision was to strike a balance between the reviewers' comments or requests for clarification and the primary relevance of these issues in relation to the manuscript.

For example the fact that negative mineral components may arise from the thermodynamic computation, even though the mass of the oxides of them mineral is always positive, is something that was not developed in this study but it is part of the design of the thermodynamic model by Ghiorso (see for example: Ghiorso and Carmichael, A Regular Solution Model for Met-

Aluminous Silicate Liquids: Applications to Geothermometry, Immiscibility, and the Source Regions of Basic Magmas, CMP, 71, 323-342, 1980; Ghiorso, A globally convergent saturation state algorithm applicable to thermodynamic systems with a stable or metastable omni-component phase, GCA, 103, 295-300, 2013).

Similarly the fact that the chemical equilibrium computations involve the solution of a constrained Gibbs free energy minimization problem is something that is not specific to this study, there are few textbooks discussing this topic (Van Zeggeren & Storey, The computation of chemical equilibrium; Smith and Missen, Chemical reaction equilibrium analysis; Sandler, Chemical and engineering thermodynamics). I have also developed several computer programs based on these principles which are completely unrelated to this study.

In the end these questions and comments were addressed in the replies to the reviewers but were only mildly addressed in the manuscript, mainly adding references to the relevant studies or books dealing with these topics. Needless to say that I am always available to provide clarifications outside the manuscript domain.

This submission includes:

- file G-KINB4.PDF revised manuscript including the supplementary material section
- file G-KINB4.DIF.PDF highlights all the changes made on the revised version of the manuscript

[revised manuscript text omitted]
_2^{2+}SiO_4} + 1\,\mathrm{Mg_3Al_2Si_3O_{12}} \Leftrightarrow 1.5\,\mathrm{Mg_2SiO_4} + 1\,\mathrm{Fe_3^{2+}Al_2Si_3O_{12}}$$

$$1\,\mathrm{Mtc} + 1\,\mathrm{OEn} \Leftrightarrow 1\,\mathrm{Fo} + 1\,\mathrm{ODi} \tag{T-2}$$

$$1\,\mathrm{CaMgSiO_4} + 1\,\mathrm{Mg2Si_2O_6} \Leftrightarrow 1\,\mathrm{Mg_2SiO_4} + 1\,\mathrm{CaMgSi_2O_6}$$

$$1\,\mathrm{Fa} + 0.5\,\mathrm{Fo} + 1\,\mathrm{OAlBff} + 1\,\mathrm{ODi} + 1\,\mathrm{OEss} \Leftrightarrow 2\,\mathrm{Mtc} + 1\,\mathrm{Alm} + 1\,\mathrm{OBff} \tag{T-3}$$

$$1\,\mathrm{Fe_2^{2+}SiO_4} + 0.5\,\mathrm{Mg_2SiO_4} + 1\,\mathrm{CaTi_{0.5}Mg_{0.5}AlSiO_6} + 1\,\mathrm{CaMgSi_2O_6} + 1\,\mathrm{CaFe^{3+}AlSiO_6} \Leftrightarrow$$
$$2\,\mathrm{CaMgSiO_4} + 1\,\mathrm{Fe_3^{2+}Al_2Si_3O_{12}} + 1\,\mathrm{CaTi_{0.5}Mg_{0.5}Fe^{3+}SiO_6}$$

$$0.5\,\mathrm{Fo} + 1\,\mathrm{OHd} \Leftrightarrow 0.5\,\mathrm{Fa} + 1\,\mathrm{ODi} \tag{T-4}$$

$$0.5\,\mathrm{Mg_2^{2+}SiO_4} + 1\,\mathrm{CaFe^{2+}Si_2O_6} \Leftrightarrow 0.5\,\mathrm{Fe_2^{2+}SiO_4} + 1\,\mathrm{CaMgSi_2O_6}$$

$$1\,\mathrm{CDi} \Leftrightarrow 1\,\mathrm{ODi} \tag{T-5}$$

$$1\,\mathrm{CaMgSi_2O_6} \Leftrightarrow 1\,\mathrm{CaMgSi_2O_6}$$

$$1\,\mathrm{Mtc} + 1\,\mathrm{CEn} \Leftrightarrow 1\,\mathrm{Fo} + 1\,\mathrm{ODi} \tag{T-6}$$

$$1\,\mathrm{CaMgSiO_4} + 1\,\mathrm{Mg2Si_2O_6} \Leftrightarrow 1\,\mathrm{Mg_2SiO_4} + 1\,\mathrm{CaMgSi_2O_6}$$

$$0.5\,\mathrm{Fo} + 1\,\mathrm{CHd} \Leftrightarrow 0.5\,\mathrm{Fa} + 1\,\mathrm{ODi} \tag{T-7}$$

$$0.5\,\mathrm{Mg_2SiO_4} + 1\,\mathrm{CaFe^{2+}Si_2O_6} \Leftrightarrow 0.5\,\mathrm{Fe_2^{2+}SiO_4} + 1\,\mathrm{CaMgSi_2O_6}$$

$$1\,\mathrm{OAlBff} \Leftrightarrow 1\,\mathrm{CAlBff} \tag{T-8}$$

$$1\,\mathrm{CaTi_{0.5}Mg_{0.5}AlSiO_6} \Leftrightarrow 1\,\mathrm{CaTi_{0.5}Mg_{0.5}AlSiO_6}$$

$$1\,\mathrm{OBff} \Leftrightarrow 1\,\mathrm{CBff} \tag{T-9}$$

$$1\,\mathrm{CaTi_{0.5}Mg_{0.5}Fe^{3+}SiO_6} \Leftrightarrow 1\,\mathrm{CaTi_{0.5}Mg_{0.5}Fe^{3+}SiO_6}$$

$$1.5\,\mathrm{Fa} + 0.5\,\mathrm{Fo} + 1\,\mathrm{ODi} + 1\,\mathrm{OAlBff} + 1\,\mathrm{CEss} \Leftrightarrow 2\,\mathrm{Mtc} + 1\,\mathrm{Alm} + 1\,\mathrm{OBff} \tag{T-10}$$

$$1.5\,\mathrm{Fe_2SiO_4} + 0.5\,\mathrm{Mg_2SiO_4} + 1\,\mathrm{CaMgSi_2O_6} + 1\,\mathrm{CaTi_{0.5}Mg_{0.5}AlSiO_6} + 1\,\mathrm{CaFe^{3+}AlSiO_6} \Leftrightarrow$$
$$2\,\mathrm{CaMgSiO_4} + 1\,\mathrm{Fe_3^{2+}Al_2Si_3O_{12}} + 1\,\mathrm{CaTi_{0.5}Mg_{0.5}Fe^{3+}SiO_6}$$

$$1\,\mathrm{CJd} \Leftrightarrow 1\,\mathrm{OJd} \tag{T-11}$$

$$1\,\mathrm{NaAlSi_2O_6} \Leftrightarrow 1\,\mathrm{NaAlSi_2O_6}$$

$$1.5\,\mathrm{Fa} + 1.5\,\mathrm{Fo} + 1\,\mathrm{Grs} \Leftrightarrow 3\,\mathrm{Mtc} + 1\,\mathrm{Alm} \tag{T-12}$$

$$1.5\,\mathrm{Fe_2^{2+}SiO_4} + 1.5\,\mathrm{Mg_2^{2+}SiO_4} + 1\,\mathrm{Ca_3Al_3Si_3O_{12}} \Leftrightarrow 3\,\mathrm{CaMgSiO_4} + 1\,\mathrm{Fe_3^{2+}Al_2Si_3O_{12}}$$

$$1\,\mathrm{Fa} + 2\,\mathrm{ODi} + 1\,\mathrm{Hc} \Leftrightarrow 2\,\mathrm{Mtc} + 1\,\mathrm{Alm} \tag{T-13}$$

$$1\,\mathrm{Fe_2^{2+}SiO_4} + 2\,\mathrm{CaMgSi_2O_6} + 1\,\mathrm{Fe^{2+}Al_2O_4} \Leftrightarrow 2\,\mathrm{CaMgSiO_4} + 1\,\mathrm{Fe_3^{2+}Al_2Si_3O_{12}}$$

$$1\,\mathrm{Fa} + 2\,\mathrm{OAlBff} + 2\,\mathrm{ODi} + 1\,\mathrm{Mag} \Leftrightarrow 2\,\mathrm{Mtc} + 1\,\mathrm{Alm} + 2\,\mathrm{OBff} \tag{T-14}$$

$$1\,\mathrm{Fe_2^{2+}SiO_4} + 2\,\mathrm{CaTi_{0.5}Mg_{0.5}AlSiO_6} + 2\,\mathrm{CaMgSi_2O_6} + 1\,\mathrm{Fe^{2+}Fe_2^{3+}O_4} \Leftrightarrow$$
$$2\,\mathrm{CaMgSiO_4} + 1\,\mathrm{Fe_3^{2+}Al_2Si_3O_{12}} + 2\,\mathrm{CaTi_{0.5}Mg_{0.5}Fe^{3+}SiO_6}$$

$$1.5\,\mathrm{Fa} + 2\,\mathrm{ODi} + 1\,\mathrm{Spl} \Leftrightarrow 2\,\mathrm{Mtc} + 0.5\,\mathrm{Fo} + 1\,\mathrm{Alm} \tag{T-15}$$

$$1.5\,\mathrm{Fe_2^{2+}SiO_4} + 2\,\mathrm{CaMgSi_2O_6} + 1\,\mathrm{MgAl_2O_4} \Leftrightarrow 2\,\mathrm{CaMgSiO_4} + 0.5\,\mathrm{Mg_2SiO_4} + 1\,\mathrm{Fe_3^{2+}Al_2Si_3O_{12}}$$

$$2\,\mathrm{Mtc} + 1\,\mathrm{Alm} + 1\,\mathrm{Ulv} \Leftrightarrow 2\,\mathrm{Fa} + 0.5\,\mathrm{Fo} + 2\,\mathrm{OAlBff} \tag{T-16}$$

$$2\,\mathrm{CaMgSiO_4} + 1\,\mathrm{Fe_3^{2+}Al_2Si_3O_{12}} + 1\,\mathrm{Fe_2^{2+}TiO_4} \Leftrightarrow 2\,\mathrm{Fe_2^{2+}SiO_4} + 0.5\,\mathrm{Mg_2^{2+}SiO_4} + 2\,\mathrm{CaTi_{0.5}Mg_{0.5}AlSiO_6}$$

[revised manuscript text omitted]

---

## Author Response (AR2)

Massimiliano Tirone
email: max.tirone@gmail.com

July 25, 2019

Executive Editor,
Solid Earth

This letter complements the submission of the revised manuscript "Chemical Heterogeneities in the Mantle: Progress Towards a General Quantitative Description" by M. Tirone (revision II).

Taking into consideration the comments of the reviewer, the manuscript has been revised to clarify few points. Some issues are only addressed in the reply.

This submission includes:

- file G-KINB5.PDF revised manuscript including the supplementary material
- file G-KINB5.DIF.PDF (added at the end of the reply) highlights all the changes made on the II revision of the manuscript

Here below I am including the reply to the reviewer and the content of the file G-KINB5.DIF.PDF.

With kind regards,

Sincerely

Massimiliano Tirone

— Reply to the Reviewer—

I praise the reviewer for taking another look at the manuscript, in fact I am actually quite intrigued by this kind of aggressive review which gives me some motivations to engage in a conversation about a study that was completed long time ago.

The reviewer is very focused on the technical/model details, but he seems to miss completely the big picture for the Earth mantle and the main point of this study. Until now the prevalent assumption is that if we have say, an eclogite and a peridotite, they are separately in chemical equilibrium and together in chemical disequilibrium, thisis the condition that at the moment defines a chemically heterogeneous mantle.
The main contribution of this study is to show that this may not be the case, we can have (partial) chemical equilibration between the two and still observe a chemically heterogeneous mantle. No need to say that the implications are quite substantial.
The equilibration model may be inaccurate, the G(*) transport a crude approximation, they both (hopefully) will be superseded by something better, while the main contribution of this study stands elsewhere and will endure.
* * *
Comment about table 3: units, sign of G(W), value of $Cr_2O_3$ (fifth column)

- Units: changed from wt% to grams where needed
- Sign of G(W): it is obviously a trivial typo
- $Cr_2O_3$ (fifth column): again not a catastrophic mistake, just a typo, 8.241 should be 8.241e-3
* * *
The "missing" experiments:
As I explained since the first submission, in a perfect scientific world, the experiments should go along side by side with a new model development. However after the funding proposal to get the first experimental data got rejected, twice by DFG and once by ERC, I can comfortably say that I won't waste any more time trying to get funded. My contribution to understand chemical heterogeneities and chemical disequilibrium in the mantle ends with this study.
* * *
Enthalpy model?
As a general rule one can use a transport equation for any energy related
quantity (with the proper terms involved) and then convert the results to
another energy quantity. The choice is usually made according to practical
considerations (e.g. convenience). For example the temperature transport
eq. used in this study and generally applied in various forms pretty much
everywhere is based on the conservation of the internal energy + kinetic
energy.

In some cases other quantities are used. For example entropy, even though
entropy is not usually conserved, which means that additional terms to de-
scribe irreversible effects must be included. Enthalpy is considered to be a
conserved quantity in magma chambers (Ghiorso & Spera docent). However
this manuscript does not deal with magma chambers or melt in general. why
enthalpy then? It turns out that I am one of the few that supported the idea
that the thermal state of the solid mantle dynamic evolution can be approx-
imated by an isentalphic model provided that the gravitational effect is also
included. In fact I am the only one who has implemented a full chemical
thermodynamic model for the mantle with this formulation, actually pub-
lished on this very same journal, Solid Earth in 2016. So why not apply it
here then?
First, it is an approximation that is valid for large scale processes when a full
geodynamic model is not available. Second, the computation is not trivial,
even less so when the local composition of the system changes with time. I
don't see the point of looking for troubles unless we have a clear evidence
that chemical equilibration among different lithologies actually happens on
a spatial scale that differ from the mantle size by few orders of magnitude or
less.

I am not aware of a geological setting where the solid mantle should be de-
scribed by the simple conservation of enthalpy formulation that the reviewer
suggested I should use.
* * *
Tables 3,4,5: To the reviewer the results may seem trivial, they weren't to
me until I actually did the modeling. I am also not sure that in certain situ-
ations with non-ideal mixtures involved, the sub-systems remain unchanged
when chemical equilibration is imposed separately. But anyway since I can-
not prove a definitive conclusion, let's share the observational results with
"ordinary" geoscientists like myself, shall we?

As for the generalization, it is hard to say, beyond what was done in section 2.1 and 2.2
* * *
The statement made by the reviewer *"if all chemical components do not diffuse at the same rate, then the conditions that arise in the diffusion couple are NOT on the binary join between the original endmembers"* is not clear at all, it does not make any sense to me. There is no binary join, there is an initial state and some boundary conditions. The boundary conditions determine the final (endmember?) states. Everything that happen inside must obey transport principles in accordance with the limits defined by the initial and boundary conditions, regardless of the diffusivity of the chemical components.

The equilibrium assemblages for the 43 cases are parameterized and then applied at the interface of the two sub-systems. Then it is possible to determine then bulk composition inside the two sub-systems because a unique relation between G(*) and the oxide abundance is also established by the 43 cases.

I added a short text on how for example one of the 43 calculations (case #11) has been set up.
* * *
Of course G is not a conserved quantity, but one can create an artificial source term at the interface (based on the results of the 43 cases).

*"And while the mass of each component may diffuse at a rate proportional to the gradient in chemical potential, the chemical potential itself does not do so (unless all activity-composition relations are linear, I guess, which they are not)."*
that is correct.

I think one major misunderstanding is to believe that the evolution model using the quantity G(*) is presented as a real exact description of the progress of the chemical equilibration process. In reality it was always intended as a semi-empirical tool aiming to highlight the existence of distinct mineralogical and compositional assemblages after chemical equilibration took place. This was also the main point of this study.
I put it down explicitly in the revised manuscript.

Quoting the reviewer *"this approach appears to enforce the unstated assumption that all components diffuse at the same rate"*
ouch! I interpret the unsupported statement about the rate of diffusion and the underline equivalence between the transport model for G(*) and chemical diffusion as an odd way to pose a certain question which I take the liberty to formulate as follow: *does the transport model for G(*) produce the same results of a multicomponent chemical diffusion model applied to the whole system assuming the intrinsic diffusion coefficients to be the same for all the bulk chemical components?*

answer: I don't know, let's find out.
I considered for comparison the 1-D output data available in the supplementary material computed with two assemblages in proportion 5:1 (figure 3 in the manuscript). The compositional variations at the interface and the size-change are known from the data file `TWOPD-G-KIN2.5.DAT`, therefore a multicomponent diffusion/growth model can be used to model the 9 oxides describing the whole assemblage on one side (I picked the left side).
Why the whole assemblage and not the individual minerals, since the composition of the minerals is available as well? Because the information about the bulk oxides for each mineral is not sufficient for a comparison with an hypothetical independent transport model applied to each mineral separately.

The diffusion matrix is constructed according to the ionic common-force model (Liang et al. GCA, 1997; Liang 2010, Rev Min & Geochim.) The self-diffusion coefficient $D_i^0$ for all ions/oxides is set initially to 0.7 (arbitrary units). This value has been chosen based on an empirical best fit assessment. For simplicity the diffusion matrix is assumed to be equivalent to the kinetic component (i.e. ideal mixing and no volumetric effect). Other details can be found in the studies referenced above here and in a Lithos paper (Tirone et al. 2016). The first figure at the end of this report shows the comparison at 3 different times, 800, 16000, 80000 (arbitrary units) (results for $Fe_2O_3$ are omitted to keep the 8 panels as large as possible fitting in one page).
Based on this figure this is my answer to the initial question: the results of the G(*) model and the multicomponent diffusion model with equal $D_i^0$ appear to be somehow similar but not quite the same.

Two additional numerical tests consider $D_{SiO2}^0$ to be 5x slower than $D_i^0$ for the rest of the oxides and 5x faster. Results are shown on the second and third figure. Little differences can be noted mainly because the mobility of the various components is interconnected.

Summary: what is the relevance of this exercise in relation to the manuscript? In my opinion it is marginally relevant. Without the experimental data, the G(*) model is as good as any other, but it has the advantage of being relatively simple and, thanks to this exercise, it seems quite general in the sense that, unless there are wild differences in the kinetic processes involved, the description provided by the G(*) model may not be too far off from the real observations This exercise confirms to me once again something that was found already in the defunct funding proposal where I made several comparison of a G(*) model with few kinetic models, considered either independently or in combination.
* * *
The text in the manuscript makes no mention of a case with f<1, *"This variation remains somehow still independent of the initial proportion of the two assemblages, at least with $f = 1, 5, 50$."*, therefore within the specified boundaries, the statement is correct.

A quantity similar to the Peclet number can be simply computed from the model setup combining the input velocity and the scaling term "S".
* * *
[Figure]

[Figure]

[Figure]

[revised manuscript text omitted]
_2^{2+}SiO_4} + 1\,\mathrm{Mg_3Al_2Si_3O_{12}} \quad \Leftrightarrow \quad 1.5\,\mathrm{Mg_2SiO_4} + 1\,\mathrm{Fe_3^{2+}Al_2Si_3O_{12}}$$

$$1\,\mathrm{Mtc} + 1\,\mathrm{OEn} \quad \Leftrightarrow \quad 1\,\mathrm{Fo} + 1\,\mathrm{ODi} \tag{T-2}$$

$$1\,\mathrm{CaMgSiO_4} + 1\,\mathrm{Mg2Si_2O_6} \quad \Leftrightarrow \quad 1\,\mathrm{Mg_2SiO_4} + 1\,\mathrm{CaMgSi_2O_6}$$

$$1\,\mathrm{Fa} + 0.5\,\mathrm{Fo} + 1\,\mathrm{OAlBff} + 1\,\mathrm{ODi} + 1\,\mathrm{OEss} \quad \Leftrightarrow \quad 2\,\mathrm{Mtc} + 1\,\mathrm{Alm} + 1\,\mathrm{OBff} \tag{T-3}$$

$$1\,\mathrm{Fe_2^{2+}SiO_4} + 0.5\,\mathrm{Mg_2SiO_4} + 1\,\mathrm{CaTi_{0.5}Mg_{0.5}AlSiO_6} \quad + \quad 1\,\mathrm{CaMgSi_2O_6} + 1\,\mathrm{CaFe^{3+}AlSiO_6} \Leftrightarrow$$

$$2\,\mathrm{CaMgSiO_4} + 1\,\mathrm{Fe_3^{2+}Al_2Si_3O_{12}} + 1\,\mathrm{CaTi_{0.5}Mg_{0.5}Fe^{3+}SiO_6}$$

$$0.5\,\mathrm{Fo} + 1\,\mathrm{OHd} \quad \Leftrightarrow \quad 0.5\,\mathrm{Fa} + 1\,\mathrm{ODi} \tag{T-4}$$

$$0.5\,\mathrm{Mg_2^{2+}SiO_4} + 1\,\mathrm{CaFe^{2+}Si_2O_6} \quad \Leftrightarrow \quad 0.5\,\mathrm{Fe_2^{2+}SiO_4} + 1\,\mathrm{CaMgSi_2O_6}$$

$$1\,\mathrm{CDi} \quad \Leftrightarrow \quad 1\,\mathrm{ODi} \tag{T-5}$$

$$1\,\mathrm{CaMgSi_2O_6} \quad \Leftrightarrow \quad 1\,\mathrm{CaMgSi_2O_6}$$

$$1\,\mathrm{Mtc} + 1\,\mathrm{CEn} \quad \Leftrightarrow \quad 1\,\mathrm{Fo} + 1\,\mathrm{ODi} \tag{T-6}$$

$$1\,\mathrm{CaMgSiO_4} + 1\,\mathrm{Mg2Si_2O_6} \quad \Leftrightarrow \quad 1\,\mathrm{Mg_2SiO_4} + 1\,\mathrm{CaMgSi_2O_6}$$

$$0.5\,\mathrm{Fo} + 1\,\mathrm{CHd} \quad \Leftrightarrow \quad 0.5\,\mathrm{Fa} + 1\,\mathrm{ODi} \tag{T-7}$$

$$0.5\,\mathrm{Mg_2SiO_4} + 1\,\mathrm{CaFe^{2+}Si_2O_6} \quad \Leftrightarrow \quad 0.5\,\mathrm{Fe_2^{2+}SiO_4} + 1\,\mathrm{CaMgSi_2O_6}$$

$$1\,\mathrm{OAlBff} \quad \Leftrightarrow \quad 1\,\mathrm{CAlBff} \tag{T-8}$$

$$1\,\mathrm{CaTi_{0.5}Mg_{0.5}AlSiO_6} \quad \Leftrightarrow \quad 1\,\mathrm{CaTi_{0.5}Mg_{0.5}AlSiO_6}$$

$$1\,\mathrm{OBff} \quad \Leftrightarrow \quad 1\,\mathrm{CBff} \tag{T-9}$$

$$1\,\mathrm{CaTi_{0.5}Mg_{0.5}Fe^{3+}SiO_6} \quad \Leftrightarrow \quad 1\,\mathrm{CaTi_{0.5}Mg_{0.5}Fe^{3+}SiO_6}$$

$$1.5\,\mathrm{Fa} + 0.5\,\mathrm{Fo} + 1\,\mathrm{ODi} + 1\,\mathrm{OAlBff} + 1\,\mathrm{CEss} \quad \Leftrightarrow \quad 2\,\mathrm{Mtc} + 1\,\mathrm{Alm} + 1\,\mathrm{OBff} \tag{T-10}$$

$$1.5\,\mathrm{Fe_2SiO_4} + 0.5\,\mathrm{Mg_2SiO_4} + 1\,\mathrm{CaMgSi_2O_6} \quad + \quad 1\,\mathrm{CaTi_{0.5}Mg_{0.5}AlSiO_6} + 1\,\mathrm{CaFe^{3+}AlSiO_6} \Leftrightarrow$$

$$2\,\mathrm{CaMgSiO_4} + 1\,\mathrm{Fe_3^{2+}Al_2Si_3O_{12}} + 1\,\mathrm{CaTi_{0.5}Mg_{0.5}Fe^{3+}SiO_6}$$

$$1\,\mathrm{CJd} \quad \Leftrightarrow \quad 1\,\mathrm{OJd} \tag{T-11}$$

$$1\,\mathrm{NaAlSi_2O_6} \quad \Leftrightarrow \quad 1\,\mathrm{NaAlSi_2O_6}$$

$$1.5\,\mathrm{Fa} + 1.5\,\mathrm{Fo} + 1\,\mathrm{Grs} \quad \Leftrightarrow \quad 3\,\mathrm{Mtc} + 1\,\mathrm{Alm} \tag{T-12}$$

$$1.5\,\mathrm{Fe_2^{2+}SiO_4} + 1.5\,\mathrm{Mg_2^{2+}SiO_4} + 1\,\mathrm{Ca_3Al_3Si_3O_{12}} \quad \Leftrightarrow \quad 3\,\mathrm{CaMgSiO_4} + 1\,\mathrm{Fe_3^{2+}Al_2Si_3O_{12}}$$

$$1\,\mathrm{Fa} + 2\,\mathrm{ODi} + 1\,\mathrm{Hc} \quad \Leftrightarrow \quad 2\,\mathrm{Mtc} + 1\,\mathrm{Alm} \tag{T-13}$$

$$1\,\mathrm{Fe_2^{2+}SiO_4} + 2\,\mathrm{CaMgSi_2O_6} + 1\,\mathrm{Fe^{2+}Al_2O_4} \quad \Leftrightarrow \quad 2\,\mathrm{CaMgSiO_4} + 1\,\mathrm{Fe_3^{2+}Al_2Si_3O_{12}}$$

$$1\,\mathrm{Fa} + 2\,\mathrm{OAlBff} + 2\,\mathrm{ODi} + 1\,\mathrm{Mag} \quad \Leftrightarrow \quad 2\,\mathrm{Mtc} + 1\,\mathrm{Alm} + 2\,\mathrm{OBff} \tag{T-14}$$

$$1\,\mathrm{Fe_2^{2+}SiO_4} + 2\,\mathrm{CaTi_{0.5}Mg_{0.5}AlSiO_6} + 2\,\mathrm{CaMgSi_2O_6} \quad + \quad 1\,\mathrm{Fe^{2+}Fe_2^{3+}O_4} \Leftrightarrow$$

$$2\,\mathrm{CaMgSiO_4} + 1\,\mathrm{Fe_3^{2+}Al_2Si_3O_{12}} + 2\,\mathrm{CaTi_{0.5}Mg_{0.5}Fe^{3+}SiO_6}$$

$$1.5\,\mathrm{Fa} + 2\,\mathrm{ODi} + 1\,\mathrm{Spl} \quad \Leftrightarrow \quad 2\,\mathrm{Mtc} + 0.5\,\mathrm{Fo} + 1\,\mathrm{Alm} \tag{T-15}$$

$$1.5\,\mathrm{Fe_2^{2+}SiO_4} + 2\,\mathrm{CaMgSi_2O_6} + 1\,\mathrm{MgAl_2O_4} \quad \Leftrightarrow \quad 2\,\mathrm{CaMgSiO_4} + 0.5\,\mathrm{Mg_2SiO_4} + 1\,\mathrm{Fe_3^{2+}Al_2Si_3O_{12}}$$

$$2\,\mathrm{Mtc} + 1\,\mathrm{Alm} + 1\,\mathrm{Ulv} \quad \Leftrightarrow \quad 2\,\mathrm{Fa} + 0.5\,\mathrm{Fo} + 2\,\mathrm{OAlBff} \tag{T-16}$$

$$2\,\mathrm{CaMgSiO_4} + 1\,\mathrm{Fe_3^{2+}Al_2Si_3O_{12}} + 1\,\mathrm{Fe_2^{2+}TiO_4} \quad \Leftrightarrow \quad 2\,\mathrm{Fe_2^{2+}SiO_4} + 0.5\,\mathrm{Mg_2^{2+}SiO_4} + 2\,\mathrm{CaTi_{0.5}Mg_{0.5}
[revised manuscript text omitted]